# Game-Theoretic Defenses for Adversarially Robust Conformal Prediction

**Rui Luo**                                                    *ruiluo@cityu.edu.hk*
*City University of Hong Kong*
**Jie Bao**                                                    *1486103897@qq.com*
*Huaiyin Institute of Technology*
**Suqun Cao**                                                  *caosuqun@hyit.edu.cn*
*Huaiyin Institute of Technology*
**Chuangyin Dang**                                            *mecdang@cityu.edu.hk*
*City University of Hong Kong*
**Zhixin Zhou**                                               *zzhou@alphabenito.com*
*Alpha Benito Research*

**Reviewed on OpenReview:** https://openreview.net/forum?id=SjsVobIlwL

## Abstract

Adversarial attacks pose major challenges to the reliability of deep learning models in safety-critical domains such as medical imaging and autonomous driving. In such high-stakes applications, providing reliable uncertainty quantification alongside adversarial robustness becomes crucial for safe deployment. Although conformal prediction can provide certain guarantees for model performance under such conditions, unknown attacks may violate the exchangeability assumption, resulting in the loss of coverage guarantees or excessively large predictive uncertainty. To address this, we propose a synergistic framework that integrates conformal prediction with game-theoretic defense strategies by modeling the adversarial interaction as a discrete, zero-sum game between attacker and defender. Our framework yields a Nash Equilibrium defense strategy, which we prove maintains valid coverage while minimizing the worst-case prediction set size against an optimal adversary operating within the defined attack space. Experimental results on CIFAR-10, CIFAR-100, and ImageNet further demonstrate that, under Nash equilibrium, defense models within our framework achieve valid coverage and minimal prediction set size. By bridging adversarial robustness and uncertainty quantification from a game-theoretic perspective, this work provides a verifiable defense paradigm for deploying safety-critical deep learning systems, particularly when adversarial distributions are unknown or dynamically evolving but contained within a known attack space. The Python code is available at https://github.com/bjbbbb/Game-Theoretic-CP.

## 1 Introduction

The reliability of deep learning technologies (Chen et al., 2025a) is facing systematic challenges posed by adversarial attacks. Such attacks, through carefully crafted subtle input perturbations, can cause medical imaging analysis models to produce fatal misdiagnoses (Ma et al., 2021) or induce autonomous driving systems (Badjie et al., 2024) to make dangerous decisions (Chen et al., 2025b), with potential risks that have transcended traditional algorithmic fault tolerance boundaries. Although adversarial training techniques have enhanced model robustness by proactively generating attack samples and have mitigated malicious input threats to some extent, their defense mechanisms still exhibit significant uncertainty and cannot provide verifiable robustness guarantees. Against this backdrop, Conformal Prediction (CP) has emerged as a promising paradigm, with its core advantages lying in distribution-free properties and robust uncertainty quantification capabilities. This data-driven approach generates statistically rigorous confidence intervals

and precisely calibrated probability estimates, demonstrating excellent performance in both regression and classification tasks, thereby providing a novel solution for constructing trustworthy deep learning systems.

However, the application of Conformal Prediction (CP) methods faces fundamental limitations, as their effectiveness strictly depends on the exchangeability assumption between training and test sets. This assumption is often difficult to satisfy in real-world application scenarios, particularly under adversarial attack conditions where attackers may employ unknown attack strategies to maliciously perturb input data. Existing research efforts primarily attempt to mitigate the impact of adversarial samples on test data distribution by reconstructing non-conformity scoring functions, but these approaches still face significant methodological bottlenecks. First, the strategy of expanding prediction set sizes to ensure statistical validity directly leads to a significant reduction in the decision utility of prediction results. Second, existing improvement schemes are mostly optimized for specific attack paradigms and struggle to guarantee coverage generalizability when confronting unknown attack vectors. Furthermore, the engineering implementation of complex non-conformity scoring functions presents technical obstacles, and existing methods exhibit systematic deficiencies in both computational complexity and cross-scenario generalization capabilities, which severely constrains the practical application value of CP methods in adversarial environments.

Therefore, developing a universal conformal prediction framework is essential. Such a framework needs to address three critical challenges: first, establishing a theoretical guarantee system that ensures arbitrary standard non-conformity scoring functions maintain preset coverage probabilities in open-world scenarios; second, optimizing uncertainty quantification strategies to minimize prediction set sizes while preserving prediction reliability; third, enabling the framework to simultaneously achieve effective coverage maintenance and prediction set size minimization when confronting arbitrary unknown attack vectors in adversarial environments.

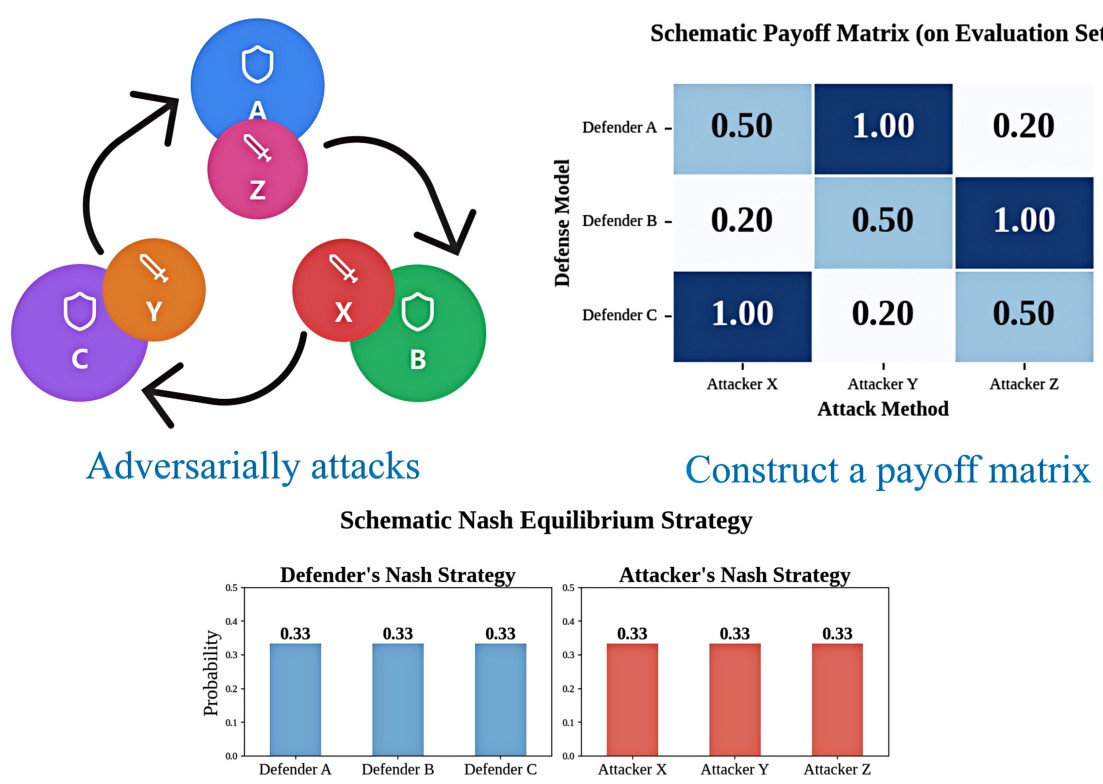

Figure 1: Address adversarial attacks in conformal prediction by constructing a game-theoretic model.

## 2 Preliminary and Problem Setup

Consider a classifier $f : \mathcal{X} \to \mathbb{R}^K$, which maps an input image $\mathrm{x} \in \mathcal{X}$ to a probability vector $f(\mathrm{x}) \in \mathbb{R}^K$. Here, $\mathcal{X} = [0,1]^d$ represents the image space, where pixel values are normalized and $d$ denotes the image dimension. The integer $K$ signifies the total number of classes, and $y \in [K] := \{1, \ldots, K\}$ denotes the ground truth label for the input $\mathrm{x}$.

### 2.1 Conformal Prediction

We first train the classifier $f$ on the training set $\mathcal{D}_{\mathrm{train}}$, and then compute nonconformity scores on the calibration set $\mathcal{D}_{\mathrm{cal}}$. On the test set $\mathcal{D}_{\mathrm{test}}$, for each input $\mathrm{x}$, we construct a prediction set $\mathcal{C}(\mathrm{x}) \subseteq [K]$ such that the true label $y$ is contained in $\mathcal{C}(\mathrm{x})$ with probability at least $1 - \alpha$, where $\alpha \in (0, 1)$ is a user-specified significance level. Formally, we guarantee

$$\mathbb{P}\left(Y \in \mathcal{C}(\mathrm{x})\right) \geq 1 - \alpha. \tag{1}$$

In conformal prediction, different nonconformity score functions $s_f(\mathrm{x}, y)$ can lead to varying outcomes, resulting in prediction sets of different sizes. For instance, a choice for the nonconformity score is $s(\mathrm{x}, k) = 1 - f_k(\mathrm{x})$, where $k$ is the class label. We compute the nonconformity scores $s(\mathrm{x}, k)$ in $\mathcal{D}_{\mathrm{cal}}$. Subsequently, the conformal quantile $\hat{q}$ is determined as the $(1 - \alpha)$-th quantile of these scores:

$$q_{1-\alpha} = \lceil (1 + |\mathcal{D}_{\mathrm{cal}}|)(1 - \alpha) \rceil \text{-th largest value in } \{s(x_i, y_i)\}_{i \in \mathcal{D}_{\mathrm{cal}}}. \tag{2}$$

This quantile provides the precise value for our prediction threshold, such that $\tau = q_{1-\alpha}$. For a new test input $\mathrm{x} \in \mathcal{D}_{\mathrm{test}}$, the prediction set $\mathcal{C}(\cdot)$ is constructed using this threshold:

$$\mathcal{C}(\mathrm{x}; f, \tau) = \{k \in [K] : s_f(\mathrm{x}, k) \leq q_{1-\alpha}\}. \tag{3}$$

This guarantee fundamentally relies on the assumption that calibration data $\mathcal{D}_{\mathrm{cal}}$ and test data $\mathcal{D}_{\mathrm{test}}$ are exchangeable.

### 2.2 Violation of Exchangeable Assumption

In adversarial scenarios, attackers manipulate the test set images $\mathcal{D}_{\mathrm{test}}$, transforming the original input samples $\mathrm{x}$ into adversarial examples $\mathrm{x}^{adv}$ through a perturbation function (attack function) $g_n$, i.e., $\mathrm{x}^{adv} = g_n(x)$. This transformation clearly disrupts the exchangeability property that underpins conformal prediction, thereby rendering standard conformal prediction unreliable in the presence of adversaries. To address this, there is a pressing need for more robust calibration and model selection strategies that explicitly account for adversarial perturbations.

### 2.3 Problem Setup

We consider a problem where an attacker selects an optimal attack $g_n^* \in \{g_n\}_{n=1}^m$ from a set of possible attacks, and a defender aims to choose a best classification model $f_j^* \in \{f_j\}_{j=1}^p$ and a threshold $\tau \in \mathbb{R}$ to minimize the expected size of the prediction set for adversarially perturbed test samples. The optimization must ensure that the true label is included in the prediction set with a probability of at least $1 - \alpha$. The attacker, within this robust framework, seeks to maximize the prediction set size. The defender formulates the problem as:

$$\min_{f_j^*, \tau \in \mathbb{R}} \max_{g_n^*} \sum_{i \in \mathcal{D}_{\mathrm{test}}} |\mathcal{C}(\mathrm{x}_i^{adv}; f_j^*, \tau)|,$$

$$\text{subject to: } \quad \mathbb{P}\left(y_i \in \mathcal{C}(\mathrm{x}_i^{adv}; f_j^*, \tau)\right) \geq 1 - \alpha. \tag{4}$$

## 3 Game-Theoretic Framework for Adversarially Robust CP

As shown in Figure 1, the optimization problem can be naturally framed as a two-player zero-sum game. As described in the problem setup, the defender aims to minimize the prediction set size while maintaining the

coverage guarantee, whereas the attacker seeks to maximize it. This section first addresses how we ensure the critical coverage constraint, $\mathbb{P}\left(y_i \in \mathcal{C}(\mathrm{x}_i^{adv}; f_j^*, \tau)\right) \geq 1 - \alpha$, under adversarial conditions, before formulating the full game.

## 3.1 Robust Coverage Guarantee under Known Adversarial Attacks

To satisfy the coverage constraint in Eq. 4 under adversarial perturbations, we employ a robust calibration strategy. The robust conformal quantile $q_{1-\alpha}^j$ for model $f_j$ is then set as the maximum across all attacks:

$$q_{1-\alpha}^j = \max_{n=1,\ldots,m} q_{1-\alpha}^{j_n} \tag{5}$$

*Remark* 1 (Intuition behind Max-Quantile). The non-conformity score quantifies the discrepancy between the prediction and the true label. Since adversarial attacks aim to induce misclassification, they typically inflate these scores compared to clean data. By selecting the maximum quantile across all considered attacks, we effectively establish a threshold calibrated to the "worst-case" perturbation. Intuitively, if a threshold is high enough to encapsulate the true label under the strongest attack (which produces the highest non-conformity scores), it will inherently cover the true label under weaker attacks, thereby preserving the coverage guarantee across the entire set of known attacks.

Prediction sets $\mathcal{C}(\mathrm{x}_i^{adv}; f_j, q_{1-\alpha}^j)$ are constructed using $q_{1-\alpha}^j$ as the threshold, consistent with Section 2.1. Additionally, we present Theorem 1, with its corresponding proof provided in Appendix A.1.

**Theorem 1** (Robust Coverage Guarantee). *For any classifier $f_j \in \{f_j\}_{j=1}^p$ and any adversarial attack $g_n \in \{g_n\}_{n=1}^m$ that was included in the calibration process used to compute $q_{1-\alpha}^j$, the prediction set $\mathcal{C}(\mathrm{x}_i^{adv}; f_j, q_{1-\alpha}^j)$ provides a coverage probability of at least $1 - \alpha$ for adversarially perturbed test samples $\mathrm{x}_i^{adv}$:*

$$\mathbb{P}\left(y_i \in \mathcal{C}(\mathrm{x}_i^{adv}; f_j, q_{1-\alpha}^j)\right) \geq 1 - \alpha. \tag{6}$$

## 3.2 Game Formulation and Nash Equilibrium

Building upon the robust coverage guarantee established in the preceding section, the constrained optimization problem presented in Eq. 4 can be formulated as a two-player zero-sum game. In this game, the defender aims to minimize the prediction set size, while the attacker seeks to maximize it.

### 3.2.1 Participants and Strategies

This game involves two participants:

- **The Defender:** The defender's strategy is to select a classifier $f_j$ from a finite set of models $\mathcal{F} = \{f_1, \ldots, f_p\}$. For each chosen $f_j$, its robust conformal quantile $q_{1-\alpha}^j$ will serve as the threshold for constructing the prediction set.

- **The Attacker:** The attacker's strategy is to select an adversarial attack $g_n$ from a finite set of known attack methods $\mathcal{G} = \{g_1, \ldots, g_m\}$.

### 3.2.2 Payoff Function

The payoff of the game is defined as the expected size of the prediction set. For a defender's chosen $f_j$ and an attacker's chosen $g_n$, the payoff (denoted as $\mathrm{Payoff}(f_j, g_n)$) is the average prediction set size on adversarially perturbed test samples:

$$\mathrm{Payoff}(f_j, g_n) = \mathbb{E}_{x \sim P_{\mathcal{X}}}\left[|\mathcal{C}(\mathbf{x}^{g_n}; f_j, q_{1-\alpha}^j)|\right] \tag{7}$$

where $P_{\mathcal{X}}$ is the distribution of the unperturbed $x$, $\mathbf{x}^{g_n}$ denotes the adversarial sample generated by applying attack $g_n$ to the original sample $\mathbf{x}$. The defender seeks to minimize this payoff, while the attacker seeks to maximize it.

---

**Algorithm 1** Game-Theoretic Framework for Adversarially Robust Conformal Prediction

---

1: **Input:** labeled data $\mathcal{D}$, unlabeled test data $\mathcal{D}_{\text{test}}$, set of defensive models $f_j, j = 1, \ldots, p$,
      set of potential attack functions $g_n, n = 1, \ldots, m$, Coverage probability $1 - \alpha$.
2: Randomly split $\mathcal{D}$ into $\{(x_i, y_i)\}_{i \in \mathcal{D}_{\text{train}}}$, $\{(x_i, y_i)\}_{i \in \mathcal{D}_{\text{cal}}}$, and $\{(x_i, y_i)\}_{i \in \mathcal{D}_{\text{eval}}}$.
    ▷ *Step 1: Train all defensive models.*
3: **for** $j = 1, \ldots, p$ **do**
4:    Train defensive model $f_j$ on the training set $\mathcal{D}_{\text{train}}$.
5: **end for**
    ▷ *Step 2: Compute robust conformal quantiles for each model.*
6: **for** each defensive model $f_j, j = 1, \ldots, p$ **do**
7:    **for** each attack method $g_n, n = 1, \ldots, m$ **do**
8:        Generate adversarial calibration set: $\mathrm{x}_i^{g_n} = g_n(\mathrm{x}_i, y_i, f_j)_{i \in \mathcal{D}_{\text{cal}}}$.
9:        Use $f_j(\mathrm{x}_i^{g_n})$ to compute non-conformity scores $\{s_j^n(\mathrm{x}_i^{g_n}, y_i)\}_{i \in \mathcal{D}_{\text{cal}}}$.
10:      Compute the quantile threshold as: $q_{1-\alpha}^{jn}$ as the $\lceil (1 + |\mathcal{D}_{\text{cal}}|)(1 - \alpha) \rceil$-th largest score in $\{s_j^n(x_i^{g_n}, y_i)\}_{i \in \mathcal{D}_{\text{cal}}}$.
11:    **end for**
12:    Determine the maximum quantile threshold $q_{1-\alpha}^j = \max_{n=1, \ldots, m} q_{1-\alpha}^{jn}$ for defensive model $f_j$.
13: **end for**
    ▷ *Step 3: Estimate the prediction set sizes on the evaluation set.*
14: **for** each defensive model $f_j, j = 1, \ldots, p$ **do**
15:    **for** each attack methods $g_n, n = 1, \ldots, m$ **do**
16:        Generate adversarial evaluation set: $\mathrm{x}_i^{g_n} = g_n(\mathrm{x}_i, y_i, f_j)_{i \in \mathcal{D}_{\text{eval}}}$.
17:        Construct the prediction set using the model's robust quantile: $\mathcal{C}(\mathrm{x}_i^{g_n}; f_j, q_{1-\alpha}^j)_{i \in \mathcal{D}_{\text{eval}}}$.
18:        Estimate the payoff by averaging the prediction set sizes over the adversarial evaluation set:
19:            $\text{Payoff}(f_j, g_n) = \frac{1}{|\mathcal{D}\text{eval}|} \sum_{i \in \mathcal{D}_{\text{eval}}} |\mathcal{C}(\mathrm{x}_i^{g_n}; f_j, q_{1-\alpha}^j)|$.
20:    **end for**
21: **end for**
    ▷ *Step 4: Find Nash Equilibrium & Construct Robust Conformal Predictor.*
22: Compute the Nash Equilibrium by solving a Linear Program: $\text{Payoff}(f_j, g_n)$.
23: Output the Nash Defense Strategy $\mathbf{d}^* = (d_1, \ldots, d_p)$ and Nash Attack Strategy $\mathbf{a}^* = (a_1, \ldots, a_m)$.
24: Use the Nash Defense Strategy $\mathbf{d}^*$ to define the final robust predictor.
    ▷ *Step 5: Testing Nash Defense Model.*
25: **for** a new test point $\mathrm{x}_{\text{new}} \in \mathcal{D}_{\text{test}}$ (which may be subjected to an unknown adversarial attack $g \in \mathcal{G}$ to become $\mathrm{x}_{\text{new}}^{adv}$) **do**
26:    Randomly sample a defensive model $f_j$ according to the Nash Equilibrium strategy $\mathbf{d}^*$.
27:    Construct and output the prediction set using the sampled model $f_j$ and its corresponding pre-computed robust quantile $q_{1-\alpha}^j$: $C(\mathrm{x}_{\text{new}}^{adv}) = \{y' : s_j(\mathrm{x}_{\text{new}}^{adv}, y') \leq q_{1-\alpha}^j\}$.
28: **end for**

---

### 3.2.3 Nash Equilibrium

Given that this is a finite two-player zero-sum game (with finite strategy sets $\mathcal{F}$ and $\mathcal{G}$), the existence of a Nash Equilibrium in mixed strategies is guaranteed by the fundamental Minimax Theorem v. Neumann (1928).

Let $\mathbf{d}$ be a mixed strategy for the defender (a probability distribution over $\mathcal{F}$) and $\mathbf{a}$ be a mixed strategy for the attacker (a probability distribution over $\mathcal{G}$). The expected payoff under these mixed strategies is $\mathbb{E}[\text{Payoff}(\mathbf{d}, \mathbf{a})]$.

**Definition (Nash Equilibrium in Mixed Strategies):** A pair of strategies $(\mathbf{d}^*, \mathbf{a}^*)$ constitutes a Nash Equilibrium if, for all other possible mixed strategies $\mathbf{d}$ and $\mathbf{a}$:

$$\mathbb{E}[\text{Payoff}(\mathbf{d}^*, \mathbf{a})] \leq \mathbb{E}[\text{Payoff}(\mathbf{d}^*, \mathbf{a}^*)] \leq \mathbb{E}[\text{Payoff}(\mathbf{d}, \mathbf{a}^*)] \tag{8}$$

In the context of our zero-sum game, the defender $\mathbf{d}$ seeks to minimize the expected payoff (prediction set size), and the attacker $\mathbf{a}$ seeks to maximize it. The Nash Equilibrium $(\mathbf{d}^*, \mathbf{a}^*)$ is the stable point where no player can gain an advantage by unilaterally deviating.

The Minimax Theorem states that the minmax value (the best the defender can do) is equal to the maxmin value (the best the attacker can do), and this common value, $V$, is the value of the game. The optimal mixed strategies $(\mathbf{d}^*, \mathbf{a}^*)$ achieve this value $V$. Formally, the value of the game $V$ is defined by the equality:

$$
\begin{aligned}
V = \mathbb{E}[\text{Payoff}(\mathbf{d}^*, \mathbf{a}^*)] &= \min_{\mathbf{d}} \max_{\mathbf{a}} \mathbb{E}[\text{Payoff}(\mathbf{d}, \mathbf{a})] \\
&= \max_{\mathbf{a}} \min_{\mathbf{d}} \mathbb{E}[\text{Payoff}(\mathbf{d}, \mathbf{a})]
\end{aligned}
\tag{9}
$$

To compute this equilibrium, we first construct a payoff matrix where rows correspond to defender's pure strategies $f_j \in \mathcal{F}$ and columns correspond to attacker's pure strategies $g_n \in \mathcal{G}$, with each entry being $\text{Payoff}(f_j, g_n)$. We then solve for the optimal mixed strategies $(\mathbf{d}^*, \mathbf{a}^*)$ by solving a Linear Program derived from the Minimax equality, which yields a solution that is robust and realistic, especially when pure strategy Nash Equilibria do not exist or when players wish to introduce unpredictability. Algorithm 1 outlines the entire process of our framework.

*Remark* 2. Our framework finds a Nash Equilibrium, defined by optimal mixed strategies for the defender $(\mathbf{d}^*)$ and attacker $(\mathbf{a}^*)$. This creates a stable state where no player can gain an advantage by unilaterally deviating, as any such move would be met by the opponent's optimal counter-play. The defender's strategy $\mathbf{d}^*$, an unpredictable probabilistic ensemble, is thus robust against the attacker's best possible strategic response, with the comprehensiveness of the attack set $\mathcal{G}$ defining the strict boundary and scope of this guarantee.

# 4 Related Work

Conformal prediction (CP) (Vovk et al., 2005) is a methodology designed to generate prediction regions for variables of interest, facilitating the estimation of model uncertainty by providing prediction sets rather than point estimates. CP has been successfully applied to both classification (Luo & Colombo, 2024; Luo & Zhou, 2025b) and regression tasks (Luo & Zhou, 2025e;f; Bao et al., 2025a; Guo et al., 2026). Its flexibility allows adaptation to various real-world scenarios, including segmentation (Luo & Zhou, 2025c), games (Luo et al., 2024), time-series forecasting (Su et al., 2024), and graph-based applications (Luo et al., 2023; Tang et al., 2025; Luo & Zhou, 2025d; Wang et al., 2025a;b; Luo & Colombo, 2025; Zhang et al., 2025).

The advent of adversarial examples (Goodfellow et al., 2014; Zhang et al., 2022) has posed substantial security challenges within the field of machine learning. In this context, uncertainty quantification emerges as a critical factor for enhancing the resilience of deep learning models. Conformal Prediction (CP) (Papadopoulos et al., 2002; Vovk et al., 2005), renowned for its ability to provide distribution-independent coverage guarantees, faces notable obstacles when confronted with data poisoning and adversarial perturbations. Empirical investigations, such as those presented by Liu et al. (2024), reveal that conventional adversarial attack strategies—including the Projected Gradient Descent (PGD) method (Madry et al., 2017)—can significantly undermine the reliability of conformal prediction frameworks.

**Adversarially Robust Conformal Prediction** To alleviate the adverse effects on conformal prediction (CP) under adversarial settings, a number of research efforts have been dedicated to tackling this challenge without relying on model retraining.

Adversarially Robust Conformal Prediction (ARCP) (Gendler et al., 2021) integrates conformal prediction with randomized smoothing to provide finite-sample coverage guarantees in the presence of $L_2$-norm-bounded adversarial noise. By incorporating Gaussian noise, it constrains the Lipschitz constant of the non-conformity score, thereby handling unknown adversarial perturbations without the need for training adjustments. Probabilistically Robust Conformal Prediction (PRCP) (Ghosh et al., 2023) employs a quantile-of-quantile approach to adapt to perturbations, establishing thresholds for both data samples and perturbations. It utilizes adversarial attacks to compute empirical robust quantiles, independently of the model training process.

Yan et al. (2024) put forward two methods, Post-Training Transformation (PTT) and Robust Conformal Training (RCT), to enhance the efficiency of robust conformal prediction. They revise RSCP into RSCP+ to provide certified guarantees and incorporate it into the training phase. Zargarbashi et al. (2024) develop robust prediction sets by bounding the worst-case variations in conformity scores under adversarial evasion and poisoning attacks. They employ CDF-based bounds to calculate conservative prediction sets and corresponding thresholds for these scenarios. Jeary et al. (2024) introduce Verifiable Robust Conformal Prediction (VRCP), which utilizes neural network verification techniques to uphold coverage guarantees amid adversarial attacks. VRCP accommodates arbitrary norm-bounded perturbations and can be extended to regression tasks.

While these approaches offer partial alleviation of the adversarial impact, they either sacrifice the compactness of the prediction sets or struggle to sustain robustness against diverse attack types and varying perturbation magnitudes.

**Adversarial Training for Conformal Prediction**  To bolster adversarial robustness in conformal prediction (CP), a straightforward and intuitive strategy involves integrating adversarial training techniques into the CP framework.

Bao et al. (2025b) put forward a method that enhances adversarial robustness under the CP setting by training on samples with uncertain attack models and incorporating conformal training with a hard threshold. Moreover, Liu et al. (2024) introduced Uncertainty-Reducing Adversarial Training (AT-UR) aimed at elevating both the efficiency and adversarial robustness of CP. This is achieved by minimizing predictive entropy and employing a weighted loss function grounded in True Class Probability Ranking. They further combined AT-UR with several established adversarial training methods, namely Adversarial Training (AT) (Madry et al., 2017), Friendly Adversarial Training (FAT) (Zhang et al., 2020), and Trade-off-inspired Adversarial Defense via Surrogate-loss Minimization (TRADES) (Zhang et al., 2019). For generating adversarial examples, they used the Projected Gradient Descent (PGD) approach.

## 5  Experiment Results

### 5.1  Experimental Setup

To comprehensively evaluate the effectiveness and robustness of our proposed game-theoretic framework across datasets of varying complexity and scale, we have designed a series of detailed experiments. This section will provide a thorough introduction to the datasets used in the experiments, the defense models, the attack methods, as well as the specific parameter configurations.

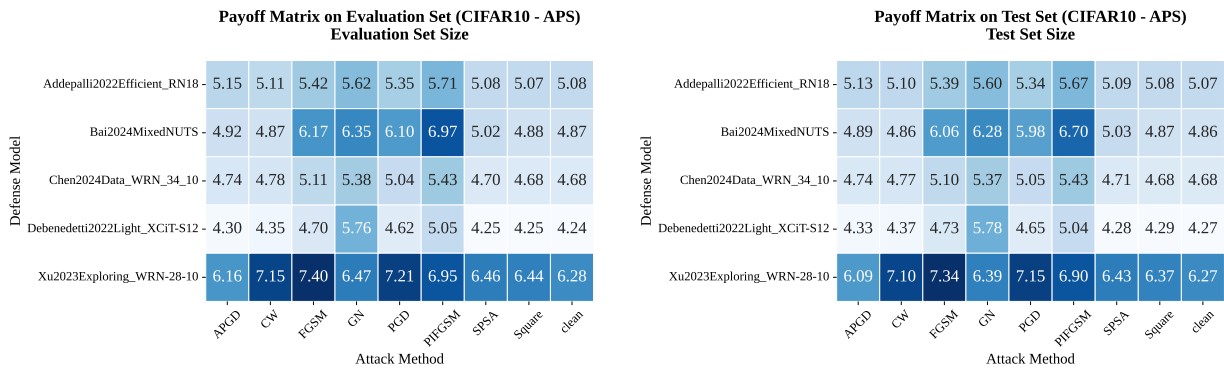

Figure 2: Evaluate the payoff matrix on both the CIFAR-10 dataset's evaluation set and test set using the APS method.

Table 1: Performance of various Conformal Prediction (CP) methods under the Nash defense strategy on the CIFAR-10 dataset. The table presents the mean and standard deviation of coverage, size, and SSCV (for specific indicator calculations refer to Appendix A.2.) against a range of attack strategies. All results are averaged over 20 independent random splits.

| Attacks | Indicator | APS | RANK | RAPS | SAPS | TOPK |
|---|---|---|---|---|---|---|
| Clean | Coverage | 0.997 (0.001) | 0.998 (0.001) | 0.997 (0.001) | 0.997 (0.001) | 0.997 (0.001) |
| | Size | 4.826 (0.092) | 5.263 (0.070) | 4.833 (0.095) | 4.666 (0.105) | 5.655 (0.093) |
| | SSCV | 0.100 (0.000) | 0.098 (0.001) | 0.100 (0.000) | 0.100 (0.000) | 0.097 (0.001) |
| FGSM | Coverage | 0.982 (0.003) | 0.982 (0.002) | 0.982 (0.003) | 0.981 (0.003) | 0.980 (0.003) |
| | Size | 5.237 (0.086) | 5.356 (0.076) | 5.241 (0.090) | 5.069 (0.096) | 5.653 (0.091) |
| | SSCV | 0.099 (0.002) | 0.082 (0.002) | 0.099 (0.000) | 0.098 (0.001) | 0.080 (0.003) |
| PGD | Coverage | 0.983 (0.003) | 0.984 (0.002) | 0.983 (0.002) | 0.982 (0.002) | 0.981 (0.003) |
| | Size | 5.189 (0.086) | 5.342 (0.075) | 5.193 (0.088) | 5.040 (0.099) | 5.655 (0.089) |
| | SSCV | 0.099 (0.002) | 0.084 (0.002) | 0.099 (0.001) | 0.098 (0.002) | 0.081 (0.003) |
| APGD | Coverage | 0.995 (0.001) | 0.989 (0.002) | 0.995 (0.001) | 0.991 (0.002) | 0.986 (0.002) |
| | Size | 4.879 (0.092) | 5.287 (0.066) | 4.887 (0.093) | 4.762 (0.103) | 5.654 (0.091) |
| | SSCV | 0.100 (0.000) | 0.089 (0.002) | 0.100 (0.001) | 0.100 (0.000) | 0.086 (0.002) |
| CW | Coverage | 0.997 (0.001) | 0.998 (0.001) | 0.997 (0.001) | 0.997 (0.001) | 0.997 (0.001) |
| | Size | 4.917 (0.090) | 5.291 (0.069) | 4.924 (0.091) | 5.124 (0.095) | **5.658 (0.089)** |
| | SSCV | 0.100 (0.000) | 0.098 (0.001) | 0.100 (0.001) | 0.100 (0.000) | 0.097 (0.001) |
| PIFGSM | Coverage | 0.898 (0.008) | 0.898 (0.007) | 0.899 (0.008) | 0.898 (0.007) | 0.900 (0.007) |
| | Size | 5.553 (0.085) | 5.416 (0.083) | 5.563 (0.088) | 5.282 (0.094) | 5.656 (0.088) |
| | SSCV | 0.099 (0.002) | 0.005 (0.005) | 0.099 (0.002) | 0.066 (0.013) | 0.006 (0.004) |
| GN | Coverage | 0.995 (0.001) | 0.989 (0.002) | 0.995 (0.001) | 0.992 (0.001) | 0.986 (0.002) |
| | Size | **5.562 (0.091)** | **5.532 (0.072)** | **5.570 (0.086)** | **5.305 (0.078)** | 5.653 (0.092) |
| | SSCV | 0.100 (0.000) | 0.089 (0.002) | 0.100 (0.000) | 0.100 (0.000) | 0.086 (0.002) |
| Square | Coverage | 0.997 (0.001) | 0.998 (0.001) | 0.997 (0.001) | 0.998 (0.001) | 0.997 (0.001) |
| | Size | 4.833 (0.088) | 5.269 (0.068) | 4.840 (0.094) | 4.756 (0.099) | 5.655 (0.088) |
| | SSCV | 0.100 (0.000) | 0.098 (0.001) | 0.100 (0.000) | 0.100 (0.000) | 0.097 (0.001) |
| SPSA | Coverage | 0.996 (0.001) | 0.998 (0.001) | 0.997 (0.001) | 0.997 (0.001) | 0.997 (0.001) |
| | Size | 4.852 (0.091) | 5.266 (0.070) | 4.858 (0.094) | 4.699 (0.107) | 5.658 (0.088) |
| | SSCV | 0.100 (0.000) | 0.098 (0.001) | 0.100 (0.001) | 0.100 (0.000) | 0.097 (0.001) |
| Nash | Coverage | 0.949 (0.013) | 0.989 (0.001) | 0.951 (0.012) | 0.968 (0.021) | 0.982 (0.029) |
| | Size | **5.558 (0.088)** | **5.532 (0.072)** | **5.566 (0.087)** | **5.304 (0.078)** | **5.657 (0.092)** |
| | SSCV | 0.099 (0.001) | 0.089 (0.001) | 0.100 (0.001) | 0.093 (0.006) | 0.083 (0.028) |

### 5.1.1 Datasets

Our experiments are conducted on three standard computer vision benchmarks: CIFAR-10, CIFAR-100, and ImageNet. For CIFAR-10 and CIFAR-100, we utilize their official test sets. For ImageNet, we use the official validation set. For each dataset, we randomly split the corresponding set into three disjoint subsets: 30% for the calibration set ($\mathcal{D}_{\mathrm{cal}}$), 30% for the evaluation set ($\mathcal{D}_{\mathrm{eval}}$), and the remaining 40% for the final test set ($\mathcal{D}_{\mathrm{test}}$).

### 5.1.2 Adversarial Attacks

We generate adversarial examples using the *torchattacks* library (Kim, 2020), with the perturbation for all attacks constrained within an $L_\infty$-norm ball of radius $\epsilon = 8/255$. As a baseline, we use original, unperturbed

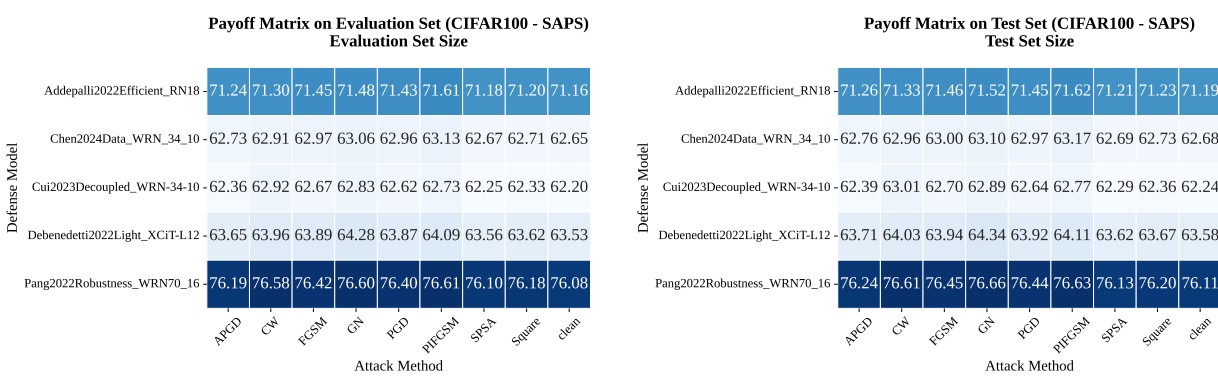

Figure 3: Evaluate the payoff matrix on both the CIFAR-100 dataset's evaluation set and test set using the SAPS method.

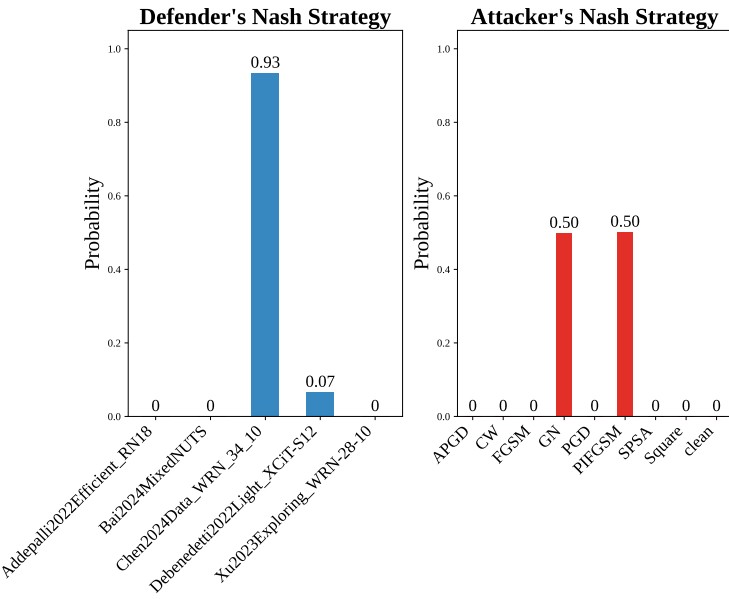

Figure 4: Obtain the Nash attack and Nash defense strategies on the CIFAR-10 dataset using the APS method.

images. The specific attack parameters are set as follows: the FGSM (Fast Gradient Sign Method) attack (Goodfellow et al., 2014) uses the default epsilon of $\epsilon = 8/255$; PGD (Projected Gradient Descent) (Madry et al., 2017) is configured with 10 iteration steps, a step size ($\alpha$) of $1/255$, and a random starting point within the $\epsilon$-ball; APGD (Auto-PGD) (Croce & Hein, 2020) is set to run for 10 steps utilizing the cross-entropy loss function. Furthermore, the CW (Carlini & Wagner) attack (Carlini & Wagner, 2017) is configured with 50 optimization steps, a learning rate of 0.01, and a confidence parameter ($c$) of 1; PIFGSM (Projected Iterative FGSM) (Gao et al., 2020) is executed for 10 iterations; and the default library implementation is used for Gaussian Noise (GN). For the CIFAR-10 and CIFAR-100 datasets, we also incorporate two additional black-box methods: the Square attack (Andriushchenko et al., 2020), which is limited to a maximum of 1000 queries per sample, and SPSA (Simultaneous Perturbation Stochastic Approximation) (Uesato et al., 2018), for which we employ the default library implementation.

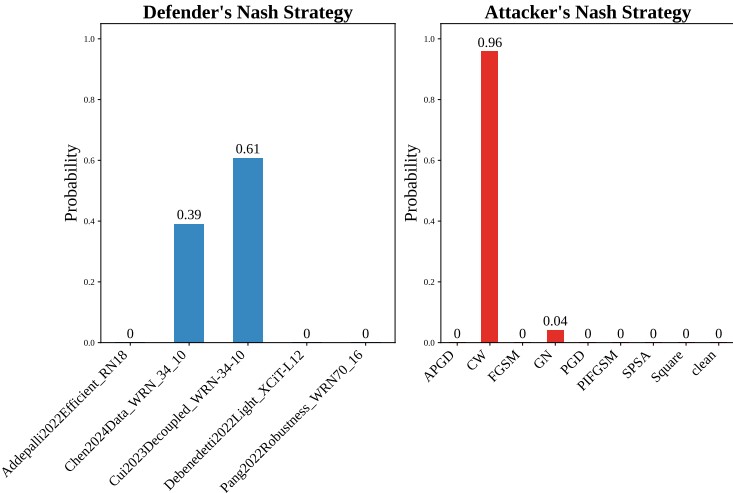

Figure 5: Obtain the Nash attack and Nash defense strategies on the CIFAR-100 dataset using the SAPS method.

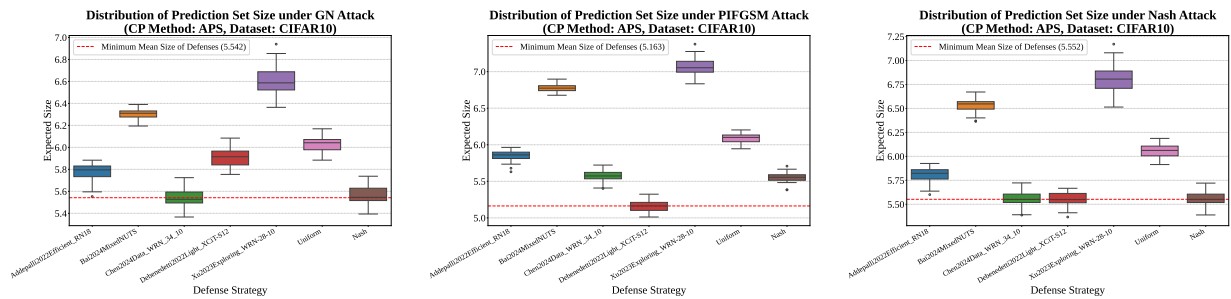

Figure 6: Box plots of size for different defense models using the APS method on the CIFAR-10 dataset under various attacks.

### 5.1.3 Defensive Models

To ensure fairness and reproducibility, all defensive models, denoted as $f_j$, are sourced from the *RobustBench benchmark* (Croce et al., 2021). RobustBench provides a standardized library of publicly available, state-of-the-art models. Since there are inconsistencies in training models across different datasets, we mainly selected these models (Addepalli et al., 2022; Bai et al., 2024; Chen & Lee, 2024; Debenedetti et al., 2023; Xu et al., 2023; Cui et al., 2024; Addepalli et al., 2022; Pang et al., 2022; Peng et al., 2023). These models have verified adversarial robustness and serve as a recognized reference for progress in the field.

In our experiments, the significance level $\alpha$ is consistently set to 0.1 across all trials. We employ five widely recognized CP methods (Huang et al., 2024a) to evaluate the performance of our framework, namely APS (Romano et al., 2020), RAPS (Angelopoulos et al., 2021), TOPK (Angelopoulos et al., 2021), SAPS (Huang et al., 2024b), and RANK (Luo & Zhou, 2025a) approaches. We use coverage, size, and SSCV (Angelopoulos et al., 2021) as evaluation metrics.

A core finding of our game-theoretic framework is the exceptional robustness of the Nash defense strategy, irrespective of the Conformal Prediction (CP) method. As shown in Table 1, our Nash defense guarantees that the expected prediction set size remains at or below the threshold set by the attacker's own Nash strategy. This upper bound holds consistently, with minor deviations attributable to the inherent generalization gap between the evaluation and test sets.

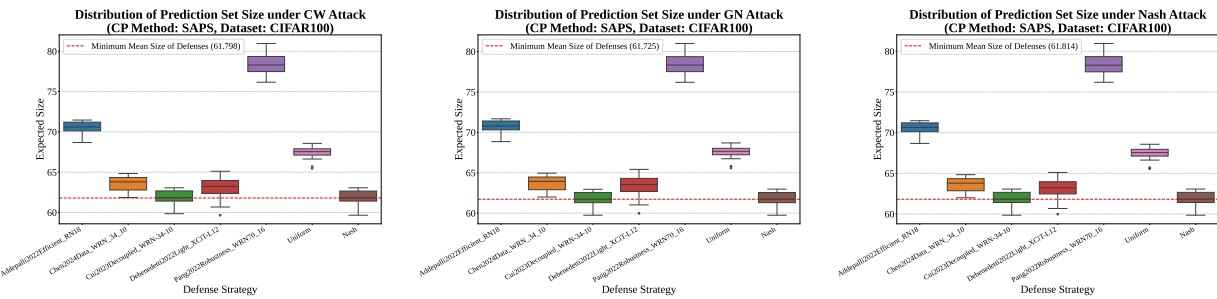

Figure 7: Box plots of size for different defense models using the SAPS method on the CIFAR-100 dataset under various attacks.

Additionally, as shown in Figures 6 and 7, when confronted with Nash attacks, our Nash defense strategy consistently outperforms many standalone defense models. While some defense models demonstrate comparable performance, their vulnerabilities are also exposed (as illustrated in the left and middle panels of Figures 6 and 7). The key advantage of Nash defense lies in its guaranteed performance ceiling: it ensures that the size of the prediction set does not exceed the upper bound defined by the Nash attack value, a guarantee that other individual models cannot provide against their respective worst-case attacks. This is the essence of its robustness: it prevents adversaries from gaining an advantage by unilaterally altering their strategies, thereby ensuring reliable worst-case performance.

As depicted in Figures 2-3, it illustrates the errors in the payoff matrices between the evaluation and test sets generated by Nash strategies, while Figures 4-5 showcases the attack and defense strategies under Nash equilibrium. We conducted detailed experiments on attack methods that fall outside the framework construction (for details, see Appendix A.3). Additionally, other results are presented in Appendix A.4-A.5.

## 6 Conclusion

In this study, we have introduced an innovative game-theoretic framework designed to tackle the significant vulnerability of conformal prediction when subjected to adversarial attacks, which fundamentally breach the foundational assumption of exchangeability. By conceptualizing the adversarial interaction between the defender and attacker as a zero-sum game, we have transitioned the defense strategy from relying on a static model selection to pursuing a dynamic, strategic equilibrium. Through rigorous theoretical analysis and extensive empirical validation on benchmark datasets such as CIFAR-10, CIFAR-100, and ImageNet, we have demonstrated that the Nash Equilibrium solution yields a mixed defense strategy that offers provable robustness within the defined game-theoretic framework. Although our Leave-One-Out experiments suggest strong empirical generalization capabilities against unseen attacks, we recognize a notable limitation: the stringent theoretical coverage guarantees are contingent upon the boundaries of the attack space. Consequently, adversarial perturbations that are exceptionally potent or markedly distinct from those in the training set may still undermine the validity of the coverage guarantees. Despite this limitation, our framework provides a verifiable defense mechanism that ensures worst-case performance against optimal adversaries within the modeled attack scope. By bridging the domains of adversarial robustness and uncertainty quantification, this work establishes a principled approach towards achieving reliable artificial intelligence. Future work could explore extending this framework to a continuous strategy space, or integrating certified robustness techniques, to provide guarantees against adversarial perturbations $g \notin G$.

## Acknowledgments

This work was supported by the National Natural Science Foundation of China (Grant 62506315) and City University of Hong Kong (Grants 9610639, 7020161).

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

# A  Appendix

## A.1  Proof for Theorem 1

*Proof.* Let $f_j$ be a given classifier and let $\mathrm{x}_i^{\mathrm{adv}}$ be a test sample generated by an arbitrary attack $g_n \in \{g_n\}_{n=1}^m$.

By the definition of the conformal quantile, for the specific attack $g_n$, the quantile $q_{1-\alpha}^{j_n}$ computed on the calibration set $\mathcal{D}_{\mathrm{cal}}$ guarantees:

$$\mathbb{P}\left(s(g_n(\mathrm{x}_i, y_i, f_j), y_i) \le q_{1-\alpha}^{j_n}\right) \ge 1 - \alpha. \tag{10}$$

The robust quantile $q_{1-\alpha}^j$ for model $f_j$ is defined as:

$$q_{1-\alpha}^j = \max_{k \in \{1,\dots,m\}} q_{1-\alpha}^{j_k}. \tag{11}$$

From this definition, it directly follows that for our chosen attack $g_n$:

$$q_{1-\alpha}^j \ge q_{1-\alpha}^{j_n}. \tag{12}$$

The event $s(\mathrm{x}_i^{\mathrm{adv}}, y_i) \le q_{1-\alpha}^{j_n}$ is a subset of the event $s(\mathrm{x}_i^{\mathrm{adv}}, y_i) \le q_{1-\alpha}^j$. Therefore, the probability of the latter is at least as large as the probability of the former:

$$\mathbb{P}\left(s(\mathrm{x}_i^{\mathrm{adv}}, y_i) \le q_{1-\alpha}^j\right) \ge \mathbb{P}\left(s(\mathrm{x}_i^{\mathrm{adv}}, y_i) \le q_{1-\alpha}^{j_n}\right). \tag{13}$$

The condition for the true label $y_i$ to be in the prediction set $\mathcal{C}(\mathrm{x}_i^{\mathrm{adv}}; f_j, q_{1-\alpha}^j)$ is precisely $s(\mathrm{x}_i^{\mathrm{adv}}, y_i) \le q_{1-\alpha}^j$. Combining these steps, we have:

$$\begin{aligned}
\mathbb{P}\left(y_i \in \mathcal{C}(\mathrm{x}_i^{\mathrm{adv}}; f_j, q_{1-\alpha}^j)\right) &= \mathbb{P}\left(s(\mathrm{x}_i^{\mathrm{adv}}, y_i) \le q_{1-\alpha}^j\right) \\
&\ge \mathbb{P}\left(s(\mathrm{x}_i^{\mathrm{adv}}, y_i) \le q_{1-\alpha}^{j_n}\right) \\
&\ge 1 - \alpha.
\end{aligned}$$

This completes the proof. $\qquad\square$

## A.2  Conformal Prediction Evaluation

The Coverage measures the proportion of test instances in $\mathcal{D}_{\mathrm{test}}$ where the true label is contained within the prediction set $\Gamma(x_i)$, and is defined as

$$\mathrm{Coverage} = \frac{1}{|\mathcal{D}_{\mathrm{test}}|} \sum_{i \in \mathcal{D}_{\mathrm{test}}} \mathbb{1}\left(y_i \in \Gamma(x_i)\right). \tag{14}$$

A higher coverage indicates that the prediction sets reliably contain the true labels.

The Size metric calculates the average number of labels in the prediction sets across all test instances,

$$\mathrm{Size} = \frac{1}{|\mathcal{D}_{\mathrm{test}}|} \sum_{i \in \mathcal{D}_{\mathrm{test}}} |\Gamma(x_i)|, \tag{15}$$

where smaller sizes denote more precise and informative predictions.

The Size-Stratified Coverage Violation (SSCV) evaluates the consistency of coverage across different prediction set sizes. It is defined as

$$\mathrm{SSCV}(\Gamma, \{S_j\}_{j=1}^s) = \sup_{j \in [s]} \left| \frac{|\{i \in J_j : y_i \in \Gamma(x_i)\}|}{|J_j|} - (1 - \alpha) \right|, \tag{16}$$

where $\{S_j\}_{j=1}^s$ partitions the possible prediction set sizes, and $J_j = \{i \in \mathcal{D}_{\mathrm{test}} : |\Gamma(x_i)| \in S_j\}$. A smaller SSCV indicates more stable coverage across different set sizes.

### A.3 Generalization to Unknown Attacks and Theoretical Discussions

In this section, we extend our analysis to evaluate the robustness of the proposed framework against unknown adversarial threats and provide further theoretical insights into the game-theoretic formulation. Specifically, we address the generalization capability of the Nash defense strategy, discuss the implications of the finite attack space assumption, and provide an intuitive interpretation of the robust coverage guarantee.

A critical consideration for any adversarial defense is its performance against attack strategies that were not encountered during the training or calibration phases. While our main framework assumes the attacker selects from a predefined set $G$, real-world scenarios may involve "unknown" attacks $g \notin G$. To rigorously evaluate the generalization capability of our Nash defense strategy, we conducted **Leave-One-Out (LOO)** and **Leave-Two-Out (LTO)** experiments.

**Experimental Setup:** In the LOO setting, we systematically exclude one attack strategy from the attacker's portfolio $G$ during the calibration of conformal quantiles and the computation of the Nash Equilibrium. The resulting defense strategy is then evaluated against the *excluded* attack. Similarly, in the LTO setting, two attacks are simultaneously excluded from the defender's knowledge base and used solely for testing.

**Analysis of Results:** The results for CIFAR-10 and CIFAR-100 are presented in Table 2 and Table 3 (referring to the tables provided in Appendix A.5 and A.6), respectively.

- **Robustness Consistency:** For the majority of attack methods, including APGD, CW, PGD, and Square Attack, the Nash defense strategy maintains valid coverage (close to or exceeding the target $1 - \alpha = 0.90$) even when these specific attacks are excluded from the calibration set. For instance, in the LOO experiments on CIFAR-10, excluding APGD results in a coverage of 0.995 against APGD itself. This suggests that the remaining attacks in $G$ provide a sufficient "basis" to approximate the worst-case boundary of the perturbation space for these attack types.

- **Limitations and Boundary Cases:** We observe a performance drop when testing against *PIFGSM* (Projected Iterative FGSM) when it is excluded from the game. As shown in the tables, the coverage against an unknown PIFGSM drops significantly (e.g., to $\sim 0.70$ in LOO settings). This highlights a fundamental property of data-driven defenses: if an unknown attack explores a region of the perturbation space that is orthogonal to or significantly more aggressive than the known set $G$, the exchangeability assumption is violated too severely for the surrogate attacks to compensate. However, when PIFGSM is included in the set $G$ (as in our main experiments), the framework successfully adapts, restoring valid coverage.

These findings demonstrate that while the framework generalizes well to unknown attacks that share structural characteristics with the known set, the defender must ensure the predefined set $G$ is diverse enough to approximate the "worst-case" capabilities of a potential adversary.

### A.4 CIFAR10 results

In this section, we provide supplementary payoff matrices (on the evaluation set) and payoff matrices (on the test set) for other methods applied to the CIFAR-10 dataset, along with corresponding results such as box plots.

We will elaborate on the specific meanings of these figures in detail. Figures 8 and 9 are box plots showing the results of using the RANK and TOPK CP methods on the CIFAR-10 dataset. The figure plots the size of the prediction sets, and it can be seen that the Nash defense yields the smallest prediction sets. Figures 10, 11, 12, and 13 respectively display the Nash attack strategies and Nash defense strategies obtained on the validation set for four different CP methods (RAPS, SAPS, RANK, TOPK). Figures 14 and 15 show the sizes of prediction sets for RAPS and SAPS as CP methods under single attacks and Nash attacks, respectively. We mainly focus on the non-single-attack scenario for Nash strategies. It can be observed that there is inconsistency among the defense models that are optimal against different single attacks. Therefore, the Nash defense employs different strategies, namely, a combined defense model. Figures 16 to 19 plot the

Table 2: Robustness against **Unknown Attacks** (Generalization) on CIFAR10. Values denote **Size (Std) / Coverage (Std)**.

| Excluded Attack(s) | APS | | RANK | | RAPS | | SAPS | | TOPK | |
|---|---|---|---|---|---|---|---|---|---|---|
| | Size | Coverage | Size | Coverage | Size | Coverage | Size | Coverage | Size | Coverage |
| **Leave-One-Out (LOO)** | | | | | | | | | | |
| Indicator | Size | Coverage | Size | Coverage | Size | Coverage | Size | Coverage | Size | Coverage |
| APGD | 4.88 (0.09) / | 0.995 (0.001) | 3.77 (0.08) / | 0.979 (0.002) | 4.89 (0.11) / | 0.995 (0.001) | 3.99 (0.19) / | 0.982 (0.003) | 3.56 (0.08) / | 0.975 (0.002) |
| CW | 4.92 (0.09) / | 0.997 (0.001) | 3.87 (0.05) / | 0.996 (0.001) | 4.93 (0.11) / | 0.997 (0.001) | 4.98 (0.09) / | 0.998 (0.001) | 3.56 (0.08) / | 0.995 (0.001) |
| FGSM | 5.23 (0.08) / | 0.981 (0.003) | 3.94 (0.03) / | 0.963 (0.002) | 5.25 (0.10) / | 0.982 (0.003) | 4.37 (0.18) / | 0.974 (0.003) | 3.56 (0.08) / | 0.955 (0.002) |
| GN | 5.89 (0.07) / | 0.996 (0.001) | 3.78 (0.07) / | 0.969 (0.003) | 5.90 (0.07) / | 0.996 (0.001) | 4.27 (0.29) / | 0.980 (0.004) | 3.57 (0.08) / | 0.962 (0.003) |
| PGD | 5.19 (0.08) / | 0.983 (0.003) | 3.91 (0.03) / | 0.970 (0.001) | 5.20 (0.11) / | 0.984 (0.003) | 4.33 (0.18) / | 0.976 (0.003) | 3.57 (0.07) / | 0.966 (0.002) |
| PIFGSM | 3.21 (0.08) / | **0.705 (0.017)** | 1.35 (0.02) / | **0.666 (0.008)** | 3.20 (0.06) / | **0.703 (0.014)** | 2.44 (0.04) / | **0.734 (0.007)** | 1.76 (0.03) / | **0.757 (0.010)** |
| SPSA | 4.86 (0.10) / | 0.997 (0.001) | 3.82 (0.07) / | 0.996 (0.001) | 4.87 (0.10) / | 0.997 (0.001) | 4.03 (0.19) / | 0.997 (0.001) | 3.54 (0.09) / | 0.995 (0.001) |
| Square | 4.84 (0.09) / | 0.997 (0.001) | 3.80 (0.07) / | 0.996 (0.001) | 4.85 (0.10) / | 0.997 (0.001) | 4.20 (0.16) / | 0.998 (0.001) | 3.54 (0.08) / | 0.995 (0.001) |
| **Leave-Two-Out (LTO)** | | | | | | | | | | |
| Indicator | Size | Coverage | Size | Coverage | Size | Coverage | Size | Coverage | Size | Coverage |
| APGD + CW | 4.92 (0.10) / | 0.996 (0.002) | 3.82 (0.08) / | 0.988 (0.009) | 4.91 (0.10) / | 0.996 (0.001) | 4.47 (0.53) / | 0.990 (0.008) | 3.59 (0.09) / | 0.985 (0.010) |
| APGD + FGSM | 5.08 (0.20) / | 0.988 (0.007) | 3.85 (0.10) / | 0.971 (0.008) | 5.06 (0.20) / | 0.988 (0.007) | 4.25 (0.31) / | 0.979 (0.005) | 3.58 (0.09) / | 0.965 (0.011) |
| APGD + GN | 5.18 (0.75) / | 0.994 (0.002) | 3.78 (0.07) / | 0.974 (0.005) | 5.17 (0.75) / | 0.994 (0.003) | 4.23 (0.38) / | 0.982 (0.004) | 3.59 (0.09) / | 0.969 (0.007) |
| APGD + PGD | 5.06 (0.18) / | 0.989 (0.006) | 3.84 (0.09) / | 0.975 (0.005) | 5.04 (0.18) / | 0.989 (0.006) | 4.23 (0.30) / | 0.980 (0.005) | 3.59 (0.09) / | 0.971 (0.005) |
| APGD + PIFGSM | 2.66 (0.43) / | **0.802 (0.101)** | 1.26 (0.06) / | **0.770 (0.120)** | 2.73 (0.44) / | **0.808 (0.105)** | 2.09 (0.25) / | **0.805 (0.088)** | 1.73 (0.04) / | **0.823 (0.076)** |
| APGD + SPSA | 4.89 (0.11) / | 0.996 (0.001) | 3.80 (0.08) / | 0.987 (0.009) | 4.87 (0.09) / | 0.996 (0.001) | 4.07 (0.25) / | 0.990 (0.008) | 3.55 (0.09) / | 0.985 (0.010) |
| APGD + Square | 4.88 (0.11) / | 0.996 (0.002) | 3.79 (0.08) / | 0.988 (0.009) | 4.86 (0.09) / | 0.996 (0.001) | 4.15 (0.25) / | 0.990 (0.008) | 3.55 (0.09) / | 0.985 (0.010) |
| CW + FGSM | 5.10 (0.18) / | 0.989 (0.008) | 3.90 (0.05) / | 0.979 (0.017) | 5.08 (0.19) / | 0.989 (0.008) | 4.66 (0.34) / | 0.986 (0.012) | 3.59 (0.09) / | 0.975 (0.020) |
| CW + GN | 5.20 (0.73) / | 0.996 (0.001) | 3.83 (0.07) / | 0.983 (0.014) | 5.20 (0.73) / | 0.996 (0.001) | 4.59 (0.41) / | 0.988 (0.010) | 3.59 (0.09) / | 0.979 (0.016) |
| CW + PGD | 5.08 (0.17) / | 0.990 (0.007) | 3.89 (0.05) / | 0.983 (0.013) | 5.06 (0.17) / | 0.990 (0.007) | 4.64 (0.36) / | 0.987 (0.012) | 3.59 (0.09) / | 0.981 (0.015) |
| CW + PIFGSM | 2.91 (0.33) / | **0.828 (0.125)** | 1.28 (0.03) / | **0.827 (0.162)** | 2.93 (0.33) / | **0.829 (0.125)** | 2.56 (0.35) / | **0.865 (0.125)** | 1.75 (0.03) / | **0.870 (0.115)** |
| CW + SPSA | 4.91 (0.11) / | 0.997 (0.001) | 3.85 (0.06) / | 0.996 (0.001) | 4.89 (0.09) / | 0.997 (0.001) | 4.48 (0.52) / | 0.998 (0.001) | 3.55 (0.09) / | 0.995 (0.001) |
| CW + Square | 4.90 (0.12) / | 0.997 (0.001) | 3.84 (0.07) / | 0.996 (0.001) | 4.88 (0.10) / | 0.997 (0.001) | 4.57 (0.43) / | 0.998 (0.001) | 3.56 (0.09) / | 0.995 (0.001) |
| FGSM + GN | 5.38 (0.55) / | 0.990 (0.007) | 3.86 (0.09) / | 0.966 (0.004) | 5.38 (0.55) / | 0.990 (0.007) | 4.42 (0.33) / | 0.978 (0.005) | 3.58 (0.09) / | 0.959 (0.005) |
| FGSM + PGD | 5.23 (0.10) / | 0.983 (0.003) | 3.92 (0.03) / | 0.966 (0.004) | 5.22 (0.10) / | 0.982 (0.003) | 4.42 (0.24) / | 0.976 (0.003) | 3.58 (0.09) / | 0.961 (0.006) |
| FGSM + PIFGSM | 2.95 (0.17) / | **0.791 (0.103)** | 1.35 (0.01) / | **0.785 (0.121)** | 2.92 (0.17) / | **0.790 (0.102)** | 2.42 (0.03) / | **0.822 (0.091)** | 1.76 (0.03) / | **0.832 (0.077)** |
| FGSM + SPSA | 5.07 (0.22) / | 0.989 (0.008) | 3.88 (0.08) / | 0.979 (0.017) | 5.04 (0.21) / | 0.989 (0.008) | 4.26 (0.29) / | 0.986 (0.011) | 3.55 (0.09) / | 0.974 (0.021) |
| FGSM + Square | 5.06 (0.23) / | 0.989 (0.008) | 3.87 (0.09) / | 0.979 (0.017) | 5.04 (0.22) / | 0.989 (0.008) | 4.34 (0.24) / | 0.987 (0.012) | 3.55 (0.09) / | 0.975 (0.021) |
| GN + PGD | 5.34 (0.59) / | 0.990 (0.006) | 3.85 (0.08) / | 0.970 (0.002) | 5.34 (0.59) / | 0.990 (0.006) | 4.40 (0.34) / | 0.979 (0.005) | 3.58 (0.09) / | 0.965 (0.003) |
| GN + PIFGSM | 3.21 (0.25) / | **0.825 (0.143)** | 1.37 (0.02) / | **0.792 (0.127)** | 3.20 (0.25) / | **0.823 (0.144)** | 2.43 (0.03) / | **0.829 (0.097)** | 1.75 (0.02) / | **0.828 (0.075)** |
| GN + SPSA | 5.15 (0.78) / | 0.996 (0.001) | 3.80 (0.07) / | 0.983 (0.013) | 5.14 (0.77) / | 0.996 (0.001) | 4.24 (0.37) / | 0.989 (0.009) | 3.55 (0.09) / | 0.978 (0.017) |
| GN + Square | 5.14 (0.78) / | 0.997 (0.001) | 3.79 (0.07) / | 0.983 (0.014) | 5.14 (0.78) / | 0.996 (0.001) | 4.32 (0.33) / | 0.990 (0.009) | 3.55 (0.09) / | 0.979 (0.017) |
| PGD + PIFGSM | 3.02 (0.20) / | **0.805 (0.104)** | 1.34 (0.02) / | **0.791 (0.127)** | 2.99 (0.20) / | **0.805 (0.103)** | 2.40 (0.05) / | **0.825 (0.092)** | 1.75 (0.03) / | **0.839 (0.085)** |
| PGD + SPSA | 5.04 (0.20) / | 0.990 (0.007) | 3.87 (0.07) / | 0.983 (0.013) | 5.02 (0.19) / | 0.990 (0.007) | 4.24 (0.28) / | 0.987 (0.011) | 3.55 (0.09) / | 0.980 (0.015) |
| PGD + Square | 5.03 (0.21) / | 0.990 (0.007) | 3.86 (0.08) / | 0.983 (0.013) | 5.01 (0.20) / | 0.990 (0.007) | 4.32 (0.23) / | 0.987 (0.011) | 3.55 (0.09) / | 0.980 (0.015) |
| PIFGSM + SPSA | 2.82 (0.39) / | **0.832 (0.129)** | 1.29 (0.06) / | **0.823 (0.159)** | 2.81 (0.38) / | **0.832 (0.130)** | 2.21 (0.23) / | **0.850 (0.118)** | 1.75 (0.03) / | **0.866 (0.112)** |
| PIFGSM + Square | 2.82 (0.40) / | **0.832 (0.129)** | 1.29 (0.06) / | **0.821 (0.157)** | 2.80 (0.39) / | **0.831 (0.130)** | 2.26 (0.19) / | **0.851 (0.118)** | 1.75 (0.03) / | **0.847 (0.093)** |
| SPSA + Square | 4.86 (0.10) / | 0.997 (0.001) | 3.81 (0.07) / | 0.996 (0.001) | 4.85 (0.10) / | 0.997 (0.001) | 4.12 (0.19) / | 0.998 (0.001) | 3.55 (0.08) / | 0.995 (0.001) |

sizes of prediction sets for four different CP methods (RANK, RAPS, SAPS, TOPK) on the evaluation set and test set under various attacks and their corresponding defense models. This is used to verify that the process is commutative. It can be seen that the sizes of prediction sets generated on the two datasets are basically consistent, meaning that our evaluation set is fully commutative with the test set. This mutually corroborates our theory of Nash defense.

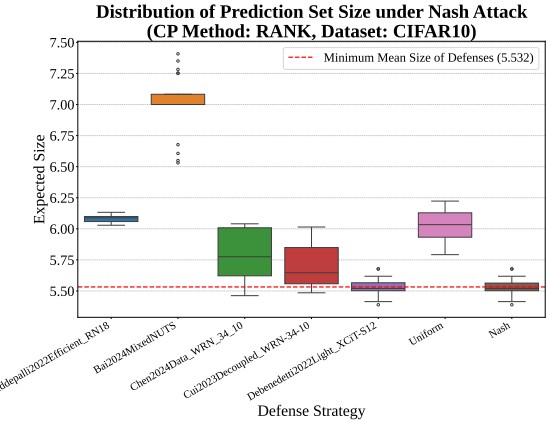

Figure 8: Box plots of size for different defense models using the RANK method on the CIFAR-10 dataset.

Table 3: Robustness against **Unknown Attacks** (Generalization) on **CIFAR100**. Values denote **Size (Std) / Coverage (Std)**.

| Excluded Attack(s) | APS | | RANK | | RAPS | | SAPS | | TOPK | |
|---|---|---|---|---|---|---|---|---|---|---|
| | Size | Coverage | Size | Coverage | Size | Coverage | Size | Coverage | Size | Coverage |
| **Leave-One-Out (LOO)** | | | | | | | | | | |
| Indicator | Size | Coverage | Size | Coverage | Size | Coverage | Size | Coverage | Size | Coverage |
| APGD | 51.94 (1.37) / 0.998 (0.001) | | 64.01 (0.96) / 0.999 (0.001) | | 52.00 (1.36) / 0.998 (0.001) | | 63.36 (0.97) / 0.999 (0.001) | | 64.26 (0.92) / 0.999 (0.001) | |
| CW | 52.28 (1.36) / 0.999 (0.001) | | 64.01 (0.96) / 0.999 (0.001) | | 52.34 (1.36) / 0.999 (0.001) | | 63.56 (0.97) / 0.999 (0.001) | | 64.26 (0.92) / 0.999 (0.001) | |
| FGSM | 54.34 (1.35) / 0.981 (0.002) | | 64.03 (0.96) / 0.981 (0.002) | | 54.40 (1.35) / 0.981 (0.002) | | 63.59 (0.98) / 0.981 (0.002) | | 64.26 (0.92) / 0.981 (0.002) | |
| GN | 57.50 (1.32) / 0.992 (0.002) | | 64.07 (0.96) / 0.990 (0.002) | | 57.55 (1.32) / 0.992 (0.002) | | 63.69 (0.97) / 0.990 (0.002) | | 64.26 (0.92) / 0.990 (0.001) | |
| PGD | 54.03 (1.36) / 0.983 (0.002) | | 64.03 (0.96) / 0.983 (0.002) | | 54.09 (1.36) / 0.983 (0.002) | | 63.57 (0.97) / 0.983 (0.002) | | 64.26 (0.92) / 0.983 (0.002) | |
| PIFGSM | 24.75 (0.77) / **0.692 (0.012)** | | 27.45 (1.13) / **0.690 (0.012)** | | 24.79 (0.89) / **0.692 (0.014)** | | 27.22 (1.13) / **0.690 (0.012)** | | 27.70 (1.12) / **0.691 (0.012)** | |
| SPSA | 51.55 (1.38) / 0.998 (0.001) | | 63.99 (0.96) / 0.998 (0.001) | | 51.51 (1.37) / 0.998 (0.001) | | 63.30 (0.93) / 0.998 (0.001) | | 64.25 (0.94) / 0.998 (0.001) | |
| Square | 51.49 (1.38) / 0.998 (0.001) | | 64.00 (0.96) / 0.999 (0.001) | | 51.45 (1.36) / 0.998 (0.001) | | 63.33 (0.93) / 0.999 (0.001) | | 64.25 (0.95) / 0.999 (0.001) | |
| **Leave-Two-Out (LTO)** | | | | | | | | | | |
| Indicator | Size | Coverage | Size | Coverage | Size | Coverage | Size | Coverage | Size | Coverage |
| APGD + CW | 52.15 (1.39) / 0.998 (0.001) | | 64.01 (0.95) / 0.999 (0.001) | | 52.18 (1.37) / 0.998 (0.001) | | 63.49 (0.97) / 0.999 (0.001) | | 64.28 (0.96) / 0.999 (0.001) | |
| APGD + FGSM | 53.18 (1.84) / 0.990 (0.009) | | 64.02 (0.95) / 0.990 (0.009) | | 53.22 (1.82) / 0.990 (0.009) | | 63.51 (0.97) / 0.990 (0.009) | | 64.28 (0.96) / 0.990 (0.009) | |
| APGD + GN | 54.76 (3.12) / 0.995 (0.004) | | 64.04 (0.95) / 0.994 (0.005) | | 54.79 (3.11) / 0.995 (0.004) | | 63.55 (0.97) / 0.994 (0.004) | | 64.28 (0.95) / 0.994 (0.005) | |
| APGD + PGD | 53.03 (1.74) / 0.991 (0.008) | | 64.02 (0.95) / 0.991 (0.008) | | 53.06 (1.72) / 0.991 (0.008) | | 63.50 (0.97) / 0.991 (0.008) | | 64.28 (0.96) / 0.991 (0.008) | |
| APGD + PIFGSM | 22.90 (2.05) / 0.830 (0.139) | | 27.42 (1.11) / 0.831 (0.143) | | 22.90 (2.07) / 0.829 (0.139) | | 27.05 (1.16) / 0.831 (0.143) | | 27.67 (1.07) / 0.831 (0.143) | |
| APGD + SPSA | 51.77 (1.38) / 0.998 (0.001) | | 64.00 (0.95) / 0.998 (0.001) | | 51.75 (1.33) / 0.998 (0.001) | | 63.38 (0.95) / 0.998 (0.001) | | 64.28 (0.98) / 0.998 (0.001) | |
| APGD + Square | 51.74 (1.38) / 0.999 (0.001) | | 64.00 (0.95) / 0.999 (0.001) | | 51.72 (1.33) / 0.998 (0.001) | | 63.39 (0.95) / 0.999 (0.001) | | 64.28 (0.98) / 0.999 (0.001) | |
| CW + FGSM | 53.35 (1.73) / 0.990 (0.009) | | 64.02 (0.95) / 0.990 (0.009) | | 53.38 (1.71) / 0.990 (0.009) | | 63.60 (0.96) / 0.990 (0.009) | | 64.28 (0.96) / 0.990 (0.009) | |
| CW + GN | 54.93 (2.97) / 0.995 (0.004) | | 64.04 (0.95) / 0.994 (0.005) | | 54.96 (2.96) / 0.995 (0.004) | | 63.65 (0.96) / 0.994 (0.004) | | 64.28 (0.96) / 0.994 (0.004) | |
| CW + PGD | 53.19 (1.64) / 0.991 (0.008) | | 64.02 (0.95) / 0.991 (0.008) | | 53.23 (1.62) / 0.991 (0.008) | | 63.59 (0.96) / 0.991 (0.008) | | 64.28 (0.96) / 0.991 (0.008) | |
| CW + PIFGSM | 23.02 (1.94) / **0.830 (0.140)** | | 27.41 (1.12) / **0.833 (0.145)** | | 23.02 (1.96) / **0.831 (0.140)** | | 27.15 (1.15) / **0.833 (0.145)** | | 27.67 (1.07) / **0.833 (0.145)** | |
| CW + SPSA | 51.94 (1.41) / 0.998 (0.001) | | 64.00 (0.95) / 0.998 (0.001) | | 51.91 (1.37) / 0.998 (0.001) | | 63.48 (0.95) / 0.998 (0.001) | | 64.29 (0.98) / 0.998 (0.001) | |
| CW + Square | 51.91 (1.42) / 0.998 (0.001) | | 64.00 (0.95) / 0.999 (0.001) | | 51.88 (1.37) / 0.998 (0.001) | | 63.49 (0.95) / 0.999 (0.001) | | 64.29 (0.98) / 0.999 (0.001) | |
| FGSM + GN | 55.96 (2.10) / 0.986 (0.006) | | 64.05 (0.95) / 0.985 (0.005) | | 55.99 (2.08) / 0.986 (0.006) | | 63.67 (0.96) / 0.985 (0.005) | | 64.28 (0.96) / 0.985 (0.005) | |
| FGSM + PGD | 54.22 (1.38) / 0.982 (0.003) | | 64.03 (0.95) / 0.982 (0.003) | | 54.26 (1.36) / 0.982 (0.002) | | 63.61 (0.96) / 0.982 (0.003) | | 64.28 (0.96) / 0.982 (0.003) | |
| FGSM + PIFGSM | 22.77 (1.20) / **0.785 (0.108)** | | 25.76 (1.04) / **0.782 (0.108)** | | 22.81 (1.19) / **0.786 (0.107)** | | 25.38 (0.98) / **0.782 (0.109)** | | 26.03 (1.00) / **0.783 (0.108)** | |
| FGSM + SPSA | 52.97 (1.97) / 0.989 (0.009) | | 64.01 (0.95) / 0.989 (0.009) | | 52.95 (1.94) / 0.989 (0.009) | | 63.49 (0.96) / 0.990 (0.009) | | 64.28 (0.98) / 0.989 (0.009) | |
| FGSM + Square | 52.94 (1.99) / 0.990 (0.009) | | 64.01 (0.95) / 0.990 (0.009) | | 52.92 (1.96) / 0.990 (0.009) | | 63.51 (0.95) / 0.990 (0.009) | | 64.28 (0.98) / 0.990 (0.009) | |
| GN + PGD | 55.80 (2.22) / 0.987 (0.005) | | 64.05 (0.95) / 0.987 (0.004) | | 55.84 (2.21) / 0.987 (0.005) | | 63.66 (0.96) / 0.987 (0.004) | | 64.28 (0.96) / 0.987 (0.004) | |
| GN + PIFGSM | 24.86 (0.84) / **0.815 (0.126)** | | 27.44 (1.12) / **0.809 (0.120)** | | 24.97 (0.73) / **0.816 (0.125)** | | 27.15 (1.11) / **0.808 (0.120)** | | 27.73 (1.07) / **0.809 (0.120)** | |
| GN + SPSA | 54.55 (3.32) / 0.995 (0.003) | | 64.03 (0.95) / 0.994 (0.004) | | 54.52 (3.30) / 0.995 (0.003) | | 63.54 (0.96) / 0.994 (0.004) | | 64.29 (0.98) / 0.994 (0.004) | |
| GN + Square | 54.52 (3.34) / 0.995 (0.004) | | 64.03 (0.95) / 0.994 (0.005) | | 54.50 (3.33) / 0.995 (0.004) | | 63.56 (0.96) / 0.994 (0.004) | | 64.29 (0.98) / 0.994 (0.005) | |
| PGD + PIFGSM | 23.63 (1.34) / **0.797 (0.108)** | | 27.43 (1.11) / **0.797 (0.109)** | | 23.74 (1.28) / **0.798 (0.107)** | | 27.09 (1.12) / **0.797 (0.109)** | | 27.73 (1.08) / **0.798 (0.108)** | |
| PGD + SPSA | 52.81 (1.86) / 0.990 (0.008) | | 64.01 (0.95) / 0.991 (0.008) | | 52.79 (1.83) / 0.990 (0.008) | | 63.48 (0.96) / 0.991 (0.008) | | 64.29 (0.98) / 0.991 (0.008) | |
| PGD + Square | 52.78 (1.88) / 0.991 (0.008) | | 64.01 (0.95) / 0.991 (0.008) | | 52.76 (1.85) / 0.991 (0.008) | | 63.50 (0.95) / 0.991 (0.008) | | 64.29 (0.98) / 0.991 (0.008) | |
| PIFGSM + SPSA | 22.48 (2.38) / **0.828 (0.139)** | | 27.40 (1.12) / **0.832 (0.143)** | | 22.59 (2.35) / **0.829 (0.138)** | | 26.96 (1.14) / **0.831 (0.143)** | | 27.72 (1.08) / **0.832 (0.143)** | |
| PIFGSM + Square | 22.50 (2.37) / **0.830 (0.140)** | | 27.40 (1.12) / **0.833 (0.145)** | | 22.59 (2.34) / **0.831 (0.140)** | | 26.97 (1.13) / **0.833 (0.145)** | | 27.72 (1.07) / **0.833 (0.145)** | |
| SPSA + Square | 51.47 (1.36) / 0.998 (0.001) | | 63.99 (0.95) / 0.998 (0.001) | | 51.52 (1.42) / 0.998 (0.001) | | 63.32 (0.88) / 0.998 (0.001) | | 64.26 (0.95) / 0.998 (0.001) | |

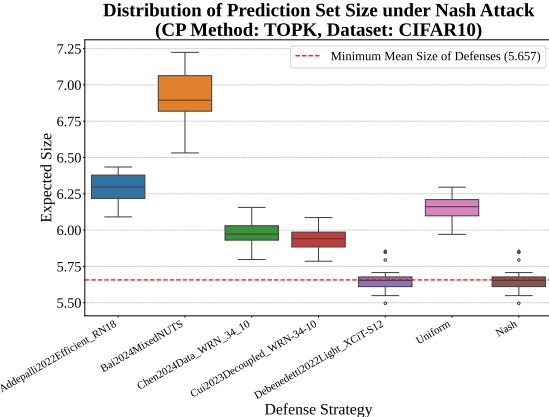

Figure 9: Box plots of size for different defense models using the TOPK method on the CIFAR-10 dataset.

## A.5 CIFAR100 results

In this section, we provide supplementary payoff matrices (on the evaluation set) and payoff matrices (on the test set) for other methods applied to the CIFAR-100 dataset, along with corresponding results such as box plots.

Figures 20 to 23 respectively present the Nash attack strategies and Nash defense strategies derived on the validation set for four different CP methods (APS, RANK, RAPS, TOPK). Figures 24 to 27 are box plots showing the results on the CIFAR-100 test set using APS, RANK, RAPS, and TOPK as CP methods. The

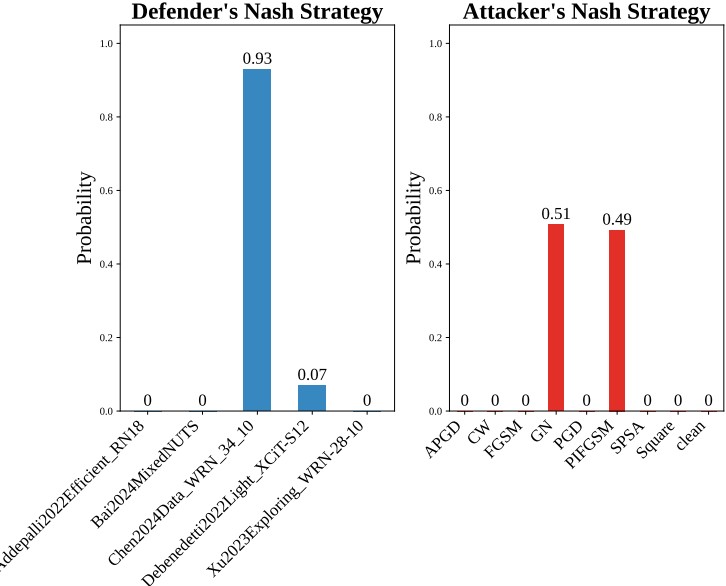

Figure 10: Obtain the Nash attack and Nash defense strategies on the CIFAR-10 dataset using the RAPS method.

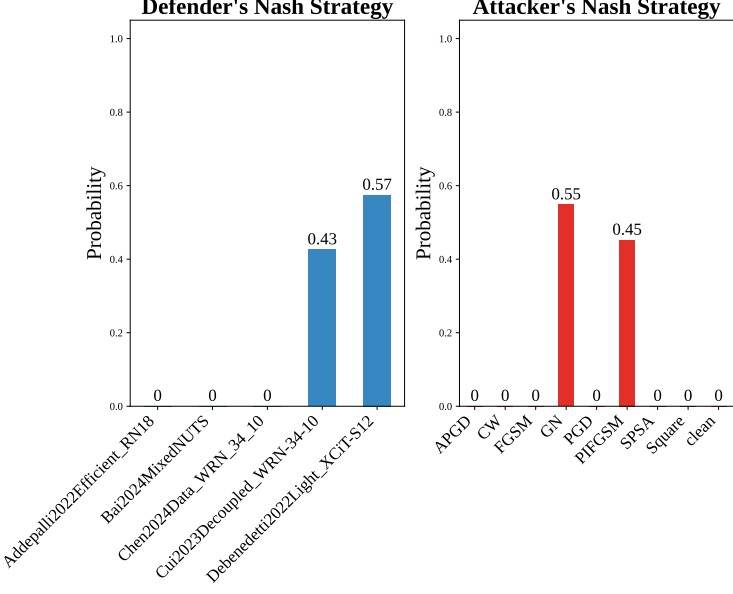

Figure 11: Obtain the Nash attack and Nash defense strategies on the CIFAR-10 dataset using the RANK method.

figures plot the sizes of prediction sets, and it can be observed that, compared to single defense models, the Nash defense generates the smallest prediction sets. When the Nash defense strategy is a single one, we plot the results for one type of attack. When the Nash attack employs a mixed strategy, we separately display

**Calculated Nash Equilibrium Strategy (CIFAR10 - RANK)**

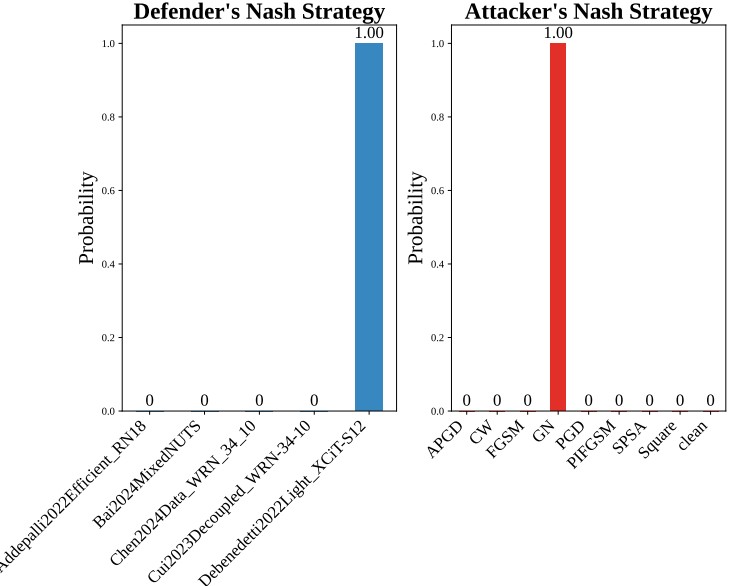

Figure 12: Obtain the Nash attack and Nash defense strategies on the CIFAR-10 dataset using the RANK method.

**Calculated Nash Equilibrium Strategy (CIFAR10 - TOPK)**

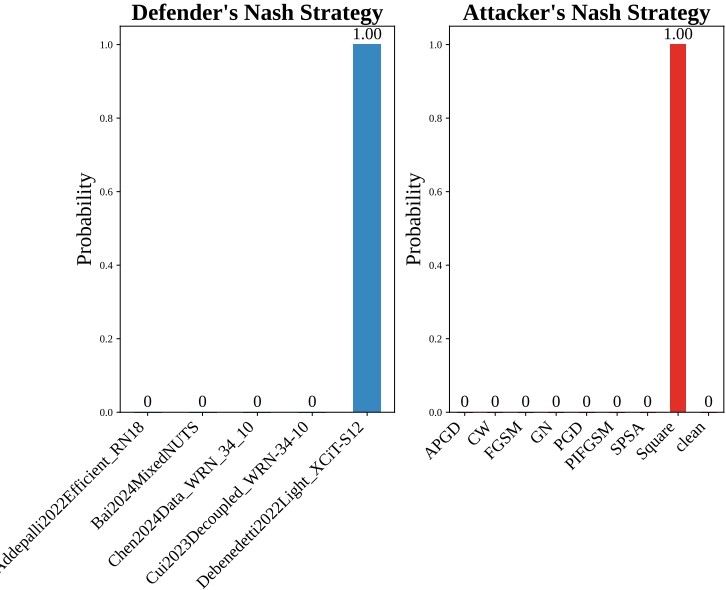

Figure 13: Obtain the Nash attack and Nash defense strategies on the CIFAR-10 dataset using the TOPK method.

box plots of the prediction set sizes for different methods under one type of attack within the mixed strategy. Figures 28 to 31 illustrate the sizes of prediction sets for four different CP methods (APS, RANK, RAPS, TOPK) on the CIFAR-100 evaluation set and test set under various attacks and their corresponding defense

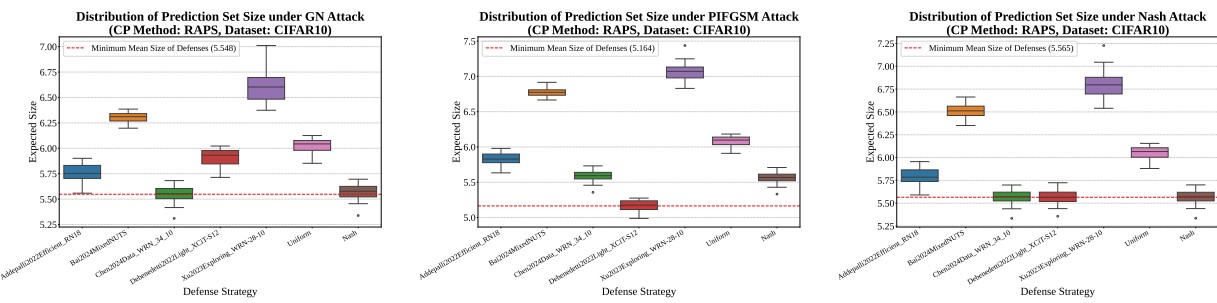

Figure 14: Box plots of size for different defense models using the RAPS method on the CIFAR-10 dataset under various attacks.

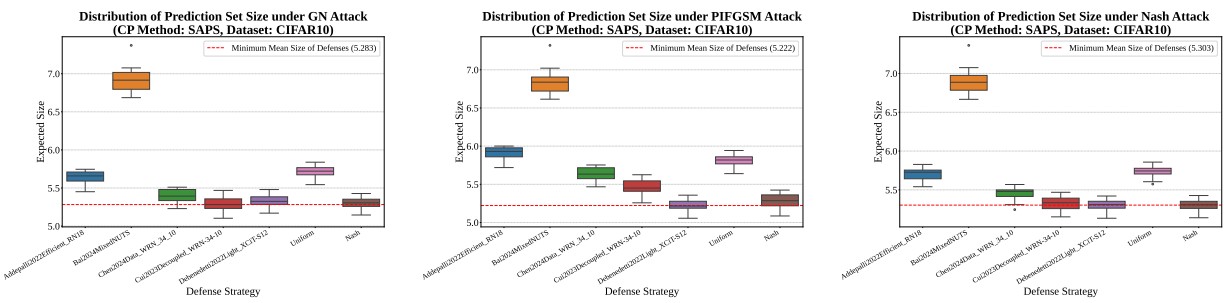

Figure 15: Box plots of size for different defense models using the SAPS method on the CIFAR-10 dataset under various attacks.

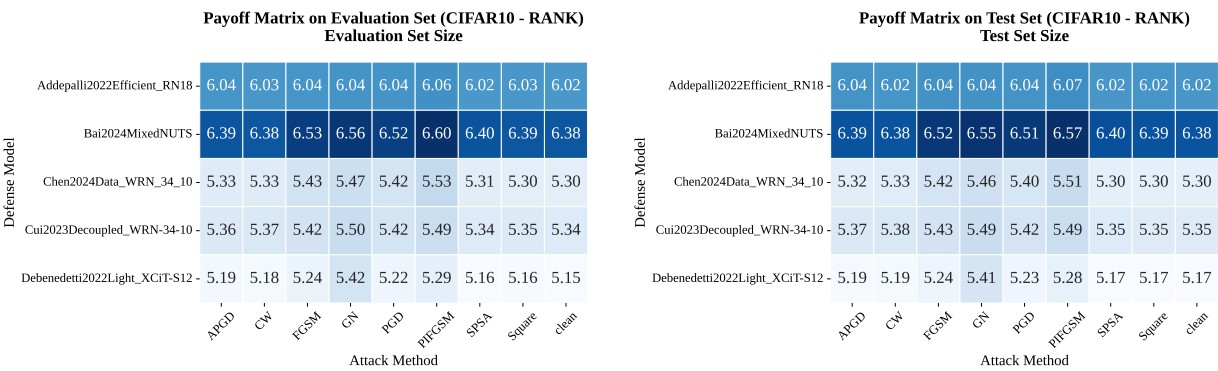

Figure 16: Evaluate the payoff matrix on both the CIFAR-10 dataset's evaluation set and test set using the RANK method.

models. This is used to verify that the process is commutative. It can be seen that the sizes of prediction sets generated on the two datasets are basically consistent, indicating that our evaluation set is fully commutative with the test set.

Table 4 presents the performance of various CP methods under the Nash defense strategy on the CIFAR-100 dataset. The table shows the average values and standard deviations of coverage, size, and SSCV for a series of attack strategies. It can be observed that under Nash attacks, the generated prediction sets are consistently the largest.

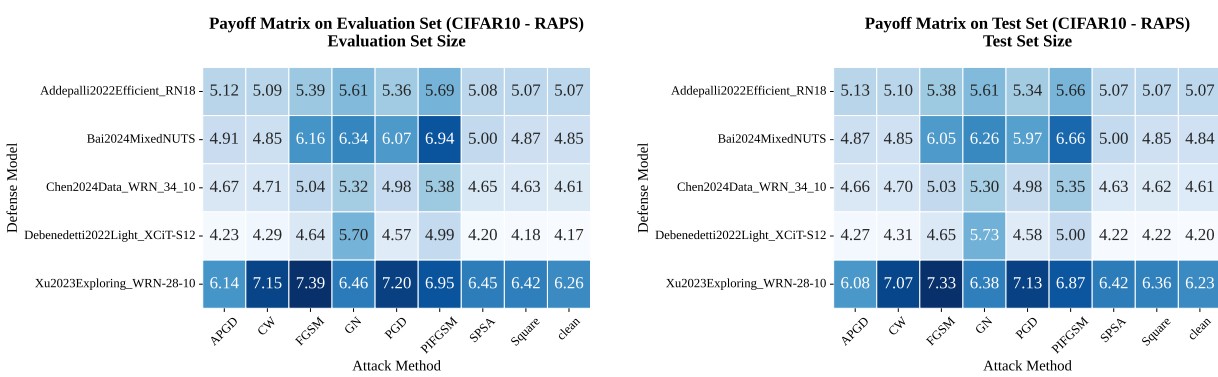

Figure 17: Evaluate the payoff matrix on both the CIFAR-10 dataset's evaluation set and test set using the RAPS method.

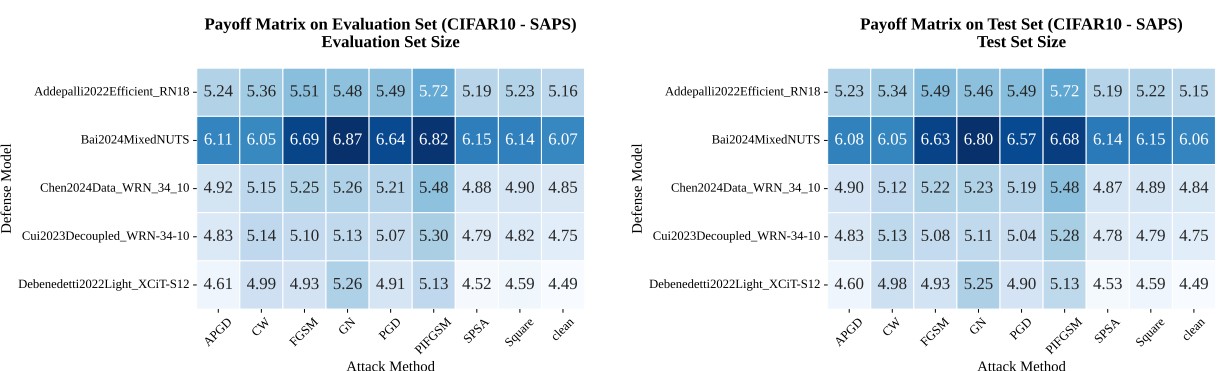

Figure 18: Evaluate the payoff matrix on both the CIFAR-10 dataset's evaluation set and test set using the SAPS method.

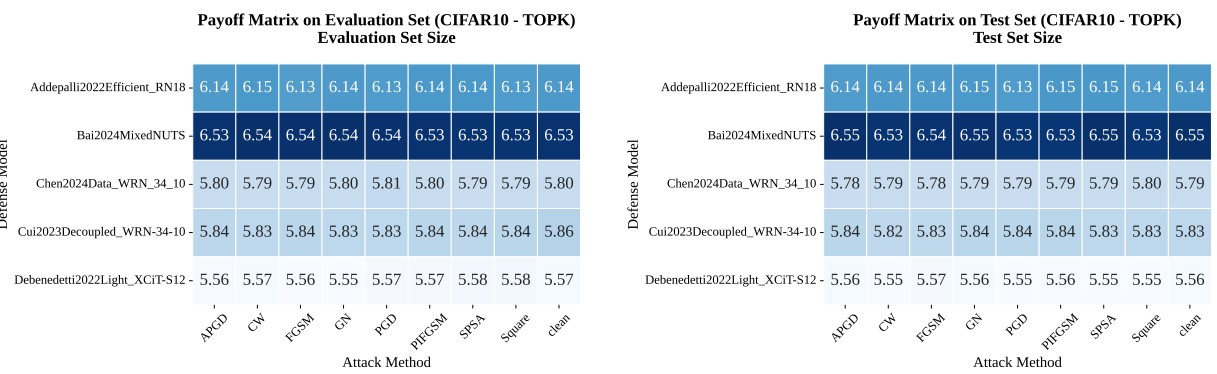

Figure 19: Evaluate the payoff matrix on both the CIFAR-10 dataset's evaluation set and test set using the TOPK method.

## A.6 ImageNet results

In this section, we provide supplementary payoff matrices (on the evaluation set) and payoff matrices (on the test set) for other methods applied to the ImageNet dataset, along with corresponding results such as box plots.

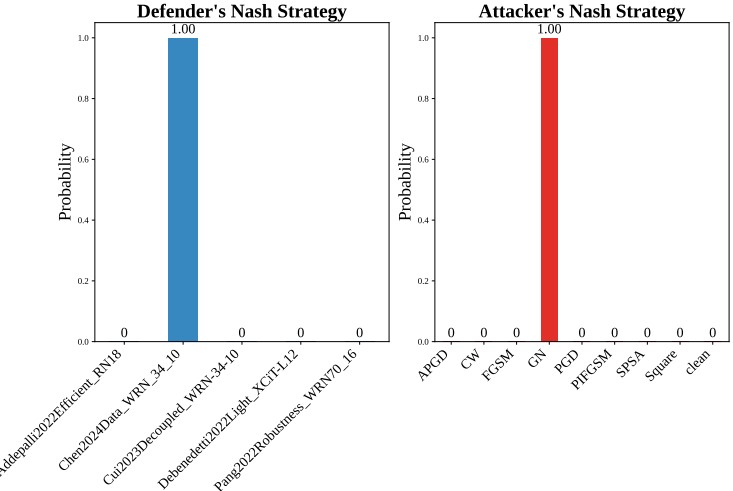

Figure 20: Obtain the Nash attack and Nash defense strategies on the CIFAR-100 dataset using the APS method.

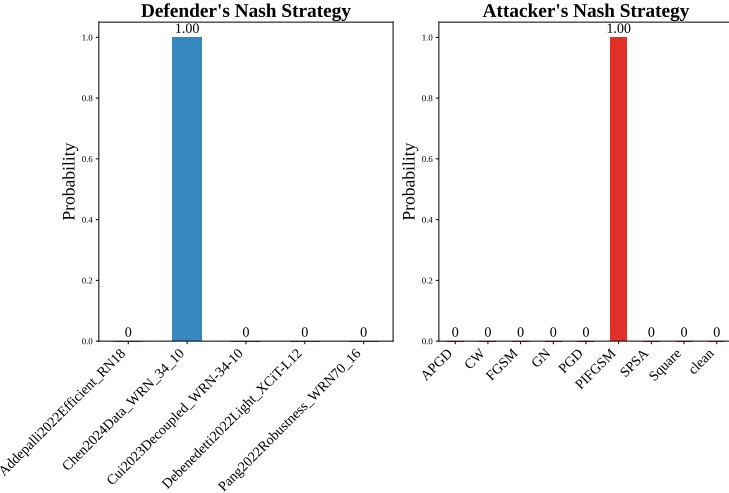

Figure 21: Obtain the Nash attack and Nash defense strategies on the CIFAR-100 dataset using the RANK method.

Figures 32 to 36 respectively present the Nash attack strategies and Nash defense strategies derived on the validation set for five different CP methods (APS, RANK, RAPS, SAPS, TOPK). Figures 37 to 41 are box plots showing the results on the ImageNet test set using APS, RANK, RAPS, SAPS, and TOPK as CP methods. Figures 42 to 46 illustrate the sizes of prediction sets for five different CP methods (APS, RANK, RAPS, SAPS, TOPK) on the evaluation set and test set under various attacks and their corresponding defense models. This is used to verify that the process is commutative. It can be observed that the sizes of prediction sets generated on the two datasets are basically consistent, indicating that our evaluation set is fully commutative with the test set.

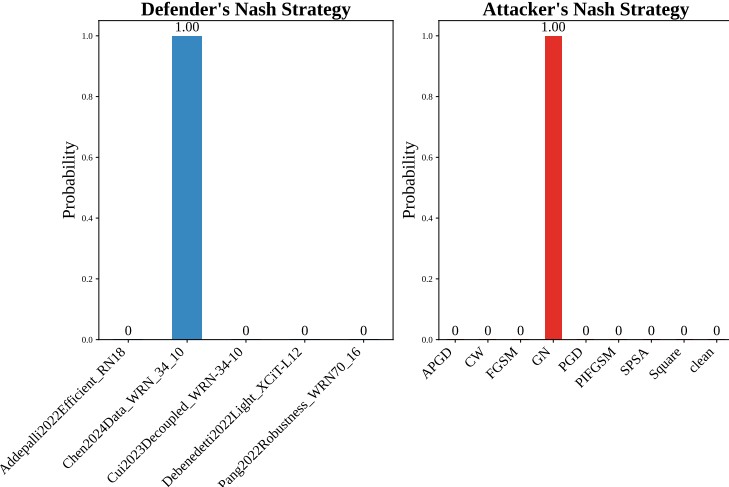

Figure 22: Obtain the Nash attack and Nash defense strategies on the CIFAR-100 dataset using the RAPS method.

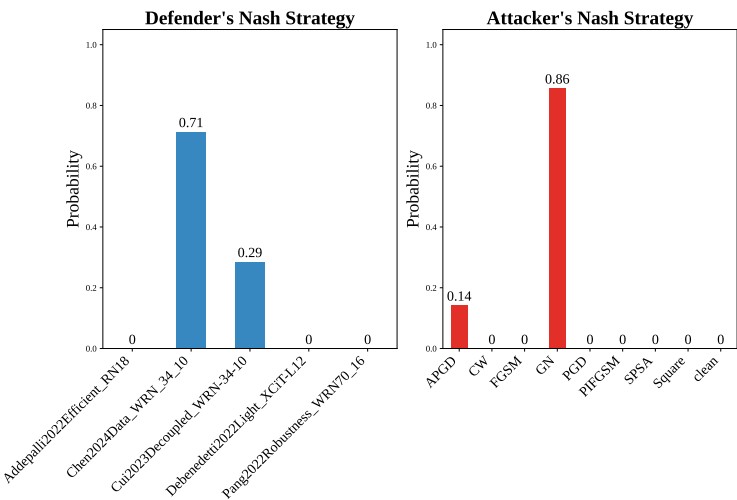

Figure 23: Obtain the Nash attack and Nash defense strategies on the CIFAR-100 dataset using the TOPK method.

Table 5 demonstrates the performance of various CP methods under the Nash defense strategy on the ImageNet dataset. The table presents the average values and standard deviations of coverage, size, and SSCV for a range of attack strategies.

Table 4: Performance of various Conformal Prediction (CP) methods under the Nash defense strategy on the CIFAR-100 dataset. The table presents the mean and standard deviation of coverage, size, and SSCV against a range of attack strategies. All results are averaged over 20 independent random splits.

| Attacks | Indicator | APS | RANK | RAPS | SAPS | TOPK |
|---|---|---|---|---|---|---|
| Clean | Coverage | 0.996 (0.003) | 0.997 (0.001) | 0.996 (0.003) | 0.997 (0.001) | 0.997 (0.001) |
| | Size | 49.198 (2.038) | 62.059 (1.023) | 48.881 (1.960) | 61.097 (0.985) | 62.304 (1.010) |
| | SSCV | 0.100 (0.001) | 0.097 (0.001) | 0.100 (0.001) | 0.097 (0.001) | 0.097 (0.001) |
| FGSM | Coverage | 0.980 (0.003) | 0.977 (0.002) | 0.979 (0.002) | 0.977 (0.002) | 0.978 (0.003) |
| | Size | 53.423 (1.117) | 62.123 (1.021) | 53.238 (1.045) | 61.535 (0.994) | 62.305 (1.005) |
| | SSCV | 0.094 (0.008) | 0.077 (0.002) | 0.093 (0.008) | 0.077 (0.002) | 0.078 (0.003) |
| PGD | Coverage | 0.981 (0.003) | 0.978 (0.002) | 0.981 (0.003) | 0.978 (0.002) | 0.979 (0.003) |
| | Size | 52.955 (1.181) | 62.113 (1.022) | 52.757 (1.106) | 61.496 (0.991) | 62.302 (1.008) |
| | SSCV | 0.095 (0.007) | 0.078 (0.002) | 0.095 (0.007) | 0.078 (0.002) | 0.079 (0.003) |
| APGD | Coverage | 0.996 (0.003) | 0.997 (0.001) | 0.996 (0.003) | 0.997 (0.001) | 0.997 (0.001) |
| | Size | 49.999 (1.845) | 62.092 (1.022) | 49.706 (1.763) | 61.233 (0.985) | 62.302 (1.010) |
| | SSCV | 0.100 (0.001) | 0.097 (0.001) | 0.100 (0.001) | 0.097 (0.001) | 0.097 (0.001) |
| CW | Coverage | 0.996 (0.003) | 0.997 (0.001) | 0.996 (0.002) | 0.997 (0.001) | 0.997 (0.001) |
| | Size | 52.307 (1.283) | 62.099 (1.020) | 52.218 (1.265) | 61.798 (1.001) | 62.303 (1.007) |
| | SSCV | 0.100 (0.001) | 0.097 (0.001) | 0.100 (0.001) | 0.097 (0.001) | 0.097 (0.001) |
| PIFGSM | Coverage | 0.898 (0.009) | 0.898 (0.006) | 0.897 (0.008) | 0.898 (0.008) | 0.896 (0.008) |
| | Size | 55.500 (1.156) | 62.133 (1.026) | 55.291 (1.099) | 61.610 (0.990) | 62.305 (1.008) |
| | SSCV | 0.143 (0.091) | 0.005 (0.004) | 0.127 (0.068) | 0.006 (0.005) | 0.007 (0.006) |
| GN | Coverage | 0.991 (0.002) | 0.988 (0.001) | 0.991 (0.002) | 0.988 (0.001) | 0.988 (0.001) |
| | Size | 57.335 (1.154) | 62.166 (1.022) | 57.225 (1.113) | 61.725 (0.970) | 62.302 (1.008) |
| | SSCV | 0.100 (0.000) | 0.088 (0.001) | 0.100 (0.000) | 0.088 (0.001) | 0.088 (0.001) |
| Square | Coverage | 0.996 (0.003) | 0.997 (0.001) | 0.996 (0.003) | 0.997 (0.001) | 0.997 (0.001) |
| | Size | 49.391 (1.940) | 62.068 (1.022) | 49.086 (1.862) | 61.198 (0.988) | 62.305 (1.010) |
| | SSCV | 0.100 (0.001) | 0.097 (0.001) | 0.100 (0.001) | 0.097 (0.001) | 0.097 (0.001) |
| SPSA | Coverage | 0.995 (0.003) | 0.996 (0.001) | 0.995 (0.003) | 0.996 (0.001) | 0.996 (0.001) |
| | Size | 49.487 (1.909) | 62.060 (1.022) | 49.184 (1.829) | 61.131 (0.986) | 62.308 (1.006) |
| | SSCV | 0.100 (0.001) | 0.096 (0.001) | 0.099 (0.001) | 0.096 (0.001) | 0.096 (0.001) |
| Nash | Coverage | 0.991 (0.002) | 0.979 (0.029) | 0.991 (0.002) | 0.996 (0.004) | 0.985 (0.024) |
| | Size | 57.335 (1.154) | 62.166 (1.022) | 57.225 (1.113) | 61.814 (0.967) | 62.302 (1.006) |
| | SSCV | 0.100 (0.000) | 0.080 (0.025) | 0.100 (0.000) | 0.096 (0.004) | 0.086 (0.019) |

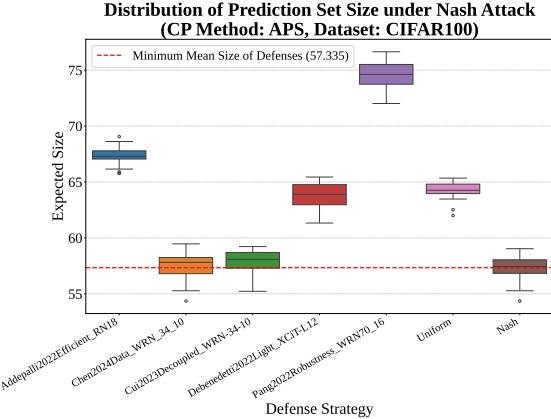

Figure 24: Box plots of size for different defense models using the APS method on the CIFAR-100 dataset.

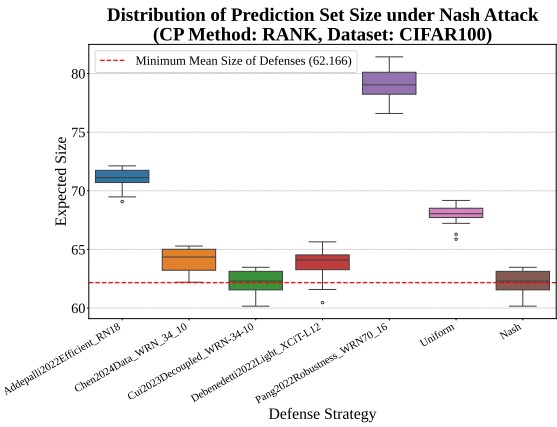

Figure 25: Box plots of size for different defense models using the RANK method on the CIFAR-100 dataset.

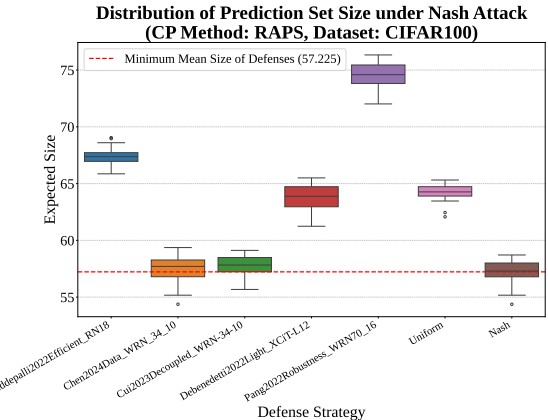

Figure 26: Box plots of size for different defense models using the RAPS method on the CIFAR-100 dataset.

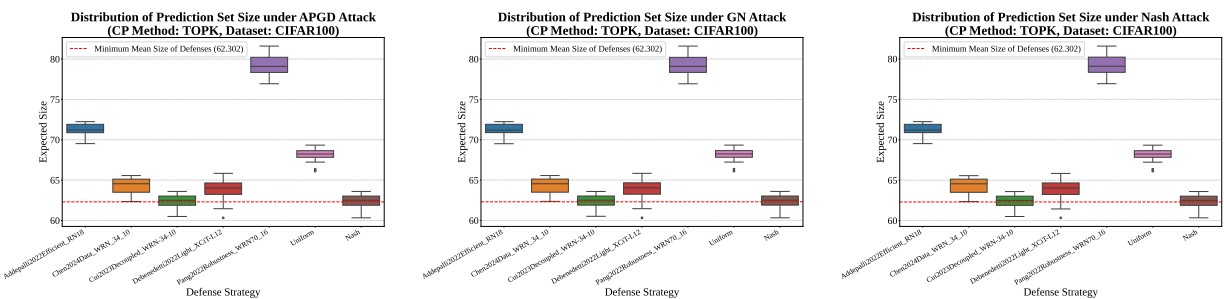

Figure 27: Box plots of size for different defense models using the TOPK method on the CIFAR-100 dataset under various attacks.

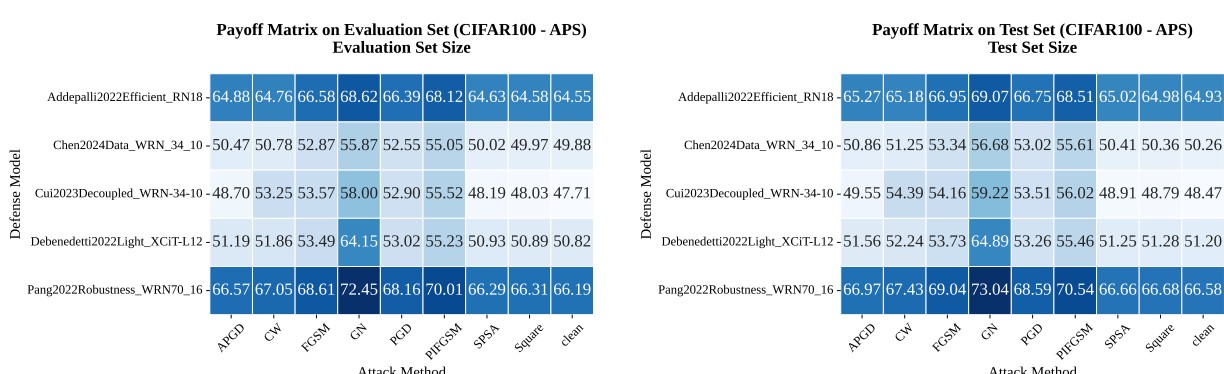

Figure 28: Evaluate the payoff matrix on both the CIFAR-100 dataset's evaluation set and test set using the APS method.

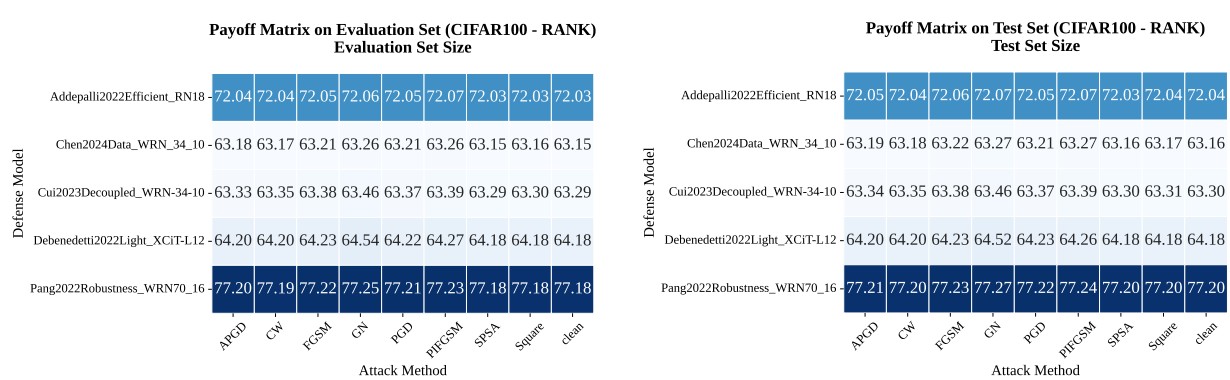

Figure 29: Evaluate the payoff matrix on both the CIFAR-100 dataset's evaluation set and test set using the RANK method.

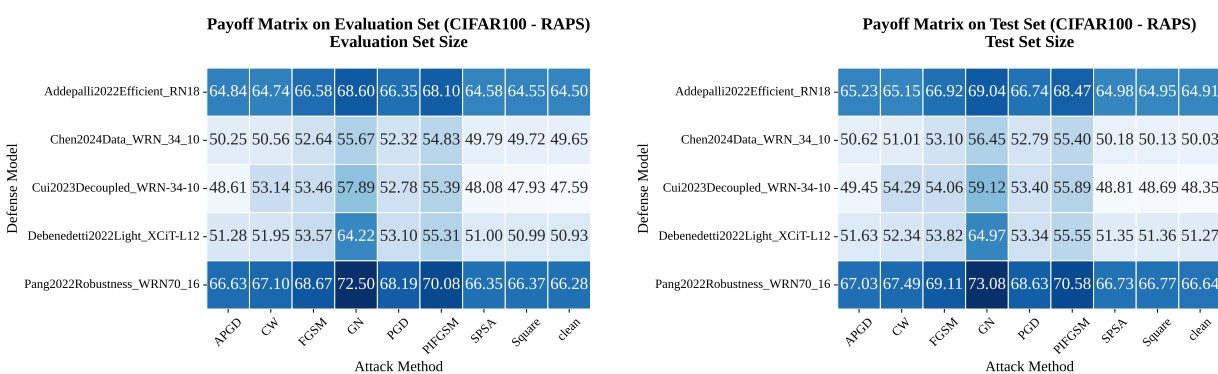

Figure 30: Evaluate the payoff matrix on both the CIFAR-100 dataset's evaluation set and test set using the RAPS method.

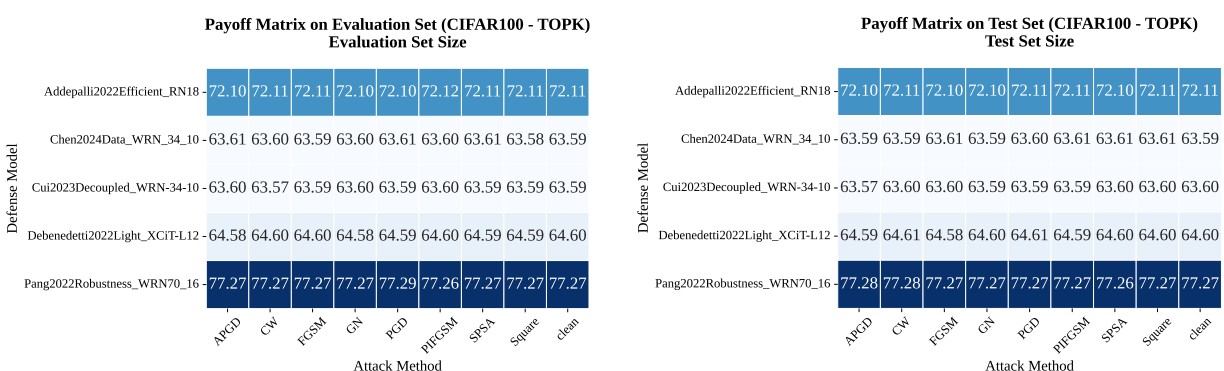

Figure 31: Evaluate the payoff matrix on both the CIFAR-100 dataset's evaluation set and test set using the TOPK method.

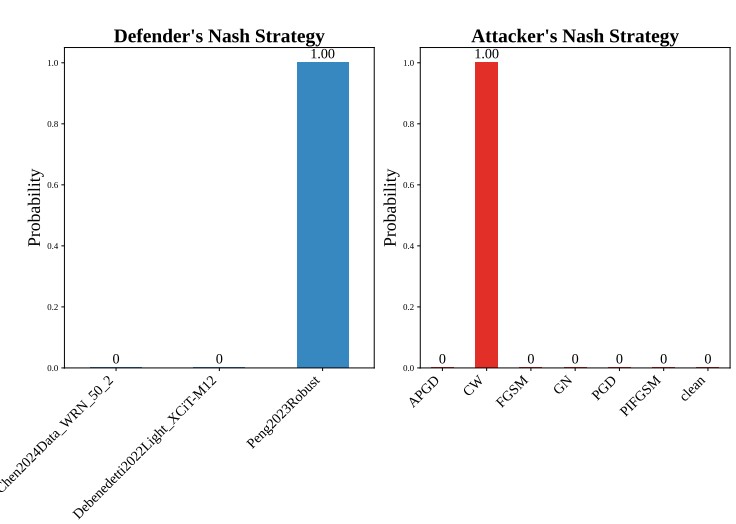

Figure 32: Obtain the Nash attack and Nash defense strategies on the ImageNet dataset using the APS method.

Table 5: Performance of various Conformal Prediction (CP) methods under the Nash defense strategy on the ImageNet dataset. The table presents the mean and standard deviation of coverage, size, and SSCV against a range of attack strategies. All results are averaged over 20 independent random splits.

| Attacks | Indicator | APS | RANK | RAPS | SAPS | TOPK |
|---|---|---|---|---|---|---|
| Clean | Coverage | 0.955 (0.001) | 0.937 (0.011) | 0.955 (0.002) | 0.938 (0.008) | 0.937 (0.011) |
| | Size | 269.403 (4.520) | 257.710 (4.929) | 269.351 (4.577) | 257.284 (4.756) | 257.565 (4.827) |
| | SSCV | 0.086 (0.003) | 0.037 (0.011) | 0.086 (0.003) | 0.038 (0.008) | 0.037 (0.011) |
| FGSM | Coverage | 0.977 (0.001) | 0.963 (0.005) | 0.977 (0.001) | 0.962 (0.004) | 0.963 (0.005) |
| | Size | 269.117 (4.453) | 257.692 (4.942) | 269.065 (4.537) | 257.111 (4.755) | 257.564 (4.826) |
| | SSCV | 0.081 (0.002) | 0.063 (0.005) | 0.081 (0.003) | 0.062 (0.004) | 0.063 (0.005) |
| PGD | Coverage | 0.964 (0.001) | 0.951 (0.003) | 0.964 (0.001) | 0.951 (0.003) | 0.951 (0.003) |
| | Size | 266.248 (4.478) | 257.689 (4.944) | 266.194 (4.567) | 257.077 (4.754) | 257.565 (4.826) |
| | SSCV | 0.072 (0.004) | 0.051 (0.003) | 0.072 (0.005) | 0.051 (0.003) | 0.051 (0.003) |
| APGD | Coverage | 0.955 (0.002) | 0.936 (0.011) | 0.955 (0.002) | 0.938 (0.008) | 0.936 (0.011) |
| | Size | 273.920 (4.421) | 257.713 (4.928) | 273.868 (4.478) | 257.329 (4.757) | 257.564 (4.826) |
| | SSCV | 0.086 (0.003) | 0.036 (0.011) | 0.086 (0.003) | 0.038 (0.008) | 0.036 (0.011) |
| CW | Coverage | 0.986 (0.001) | 0.974 (0.004) | 0.986 (0.001) | 0.974 (0.003) | 0.974 (0.004) |
| | Size | 278.106 (4.541) | 257.706 (4.934) | 278.054 (4.597) | 257.318 (4.756) | 257.565 (4.826) |
| | SSCV | 0.090 (0.003) | 0.074 (0.004) | 0.090 (0.003) | 0.074 (0.003) | 0.074 (0.004) |
| PIFGSM | Coverage | 0.901 (0.003) | 0.900 (0.003) | 0.901 (0.003) | 0.900 (0.003) | 0.900 (0.003) |
| | Size | 271.377 (4.892) | 257.689 (4.944) | 271.325 (5.014) | 257.065 (4.752) | 257.566 (4.827) |
| | SSCV | 0.138 (0.017) | 0.003 (0.002) | 0.143 (0.019) | 0.002 (0.001) | 0.003 (0.002) |
| GN | Coverage | 0.987 (0.001) | 0.976 (0.003) | 0.987 (0.001) | 0.975 (0.002) | 0.975 (0.003) |
| | Size | 253.404 (4.189) | 257.686 (4.947) | 253.355 (4.246) | 257.015 (4.757) | 257.566 (4.826) |
| | SSCV | 0.092 (0.002) | 0.076 (0.003) | 0.092 (0.002) | 0.075 (0.002) | 0.075 (0.003) |
| Nash | Coverage | 0.986 (0.001) | 0.942 (0.018) | 0.986 (0.001) | 0.949 (0.020) | 0.944 (0.026) |
| | Size | 278.106 (4.541) | 257.715 (4.929) | 278.054 (4.597) | 257.328 (4.758) | 257.565 (4.827) |
| | SSCV | 0.090 (0.003) | 0.042 (0.018) | 0.090 (0.003) | 0.049 (0.020) | 0.044 (0.026) |

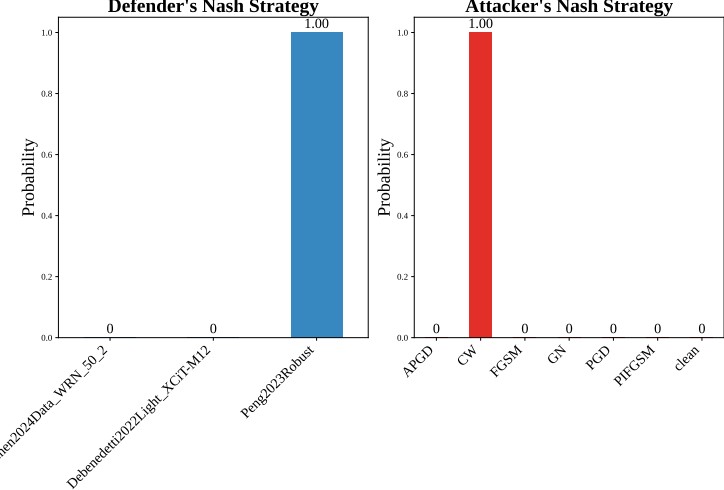

Figure 33: Obtain the Nash attack and Nash defense strategies on the ImageNet dataset using the RANK method.

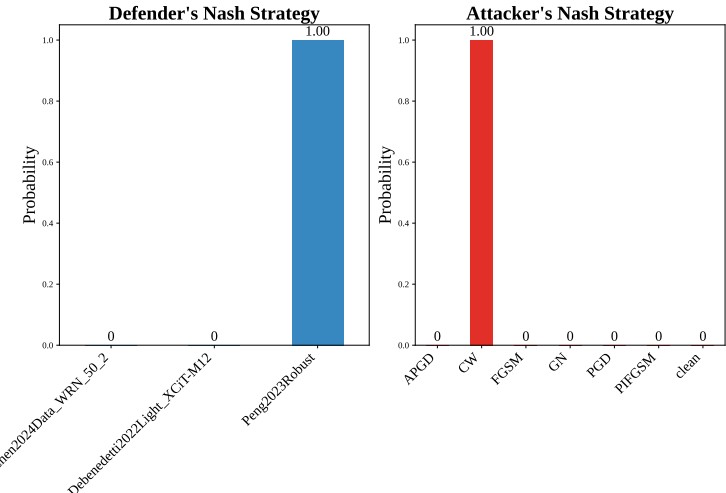

Figure 34: Obtain the Nash attack and Nash defense strategies on the ImageNet dataset using the RAPS method.

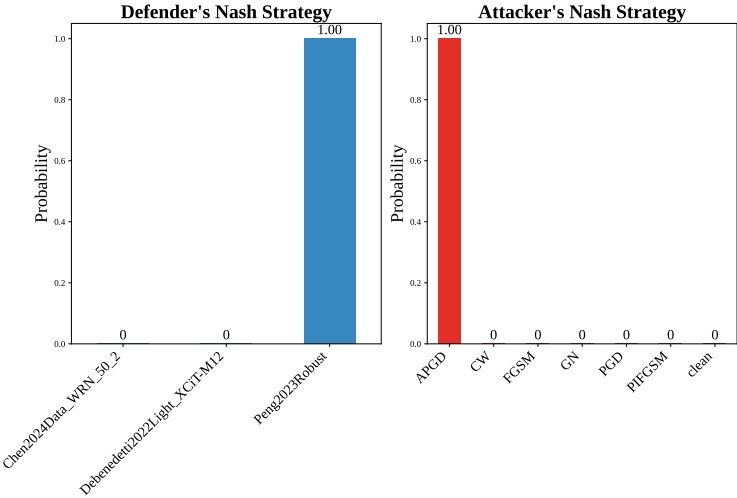

Figure 35: Obtain the Nash attack and Nash defense strategies on the ImageNet dataset using the SAPS method.

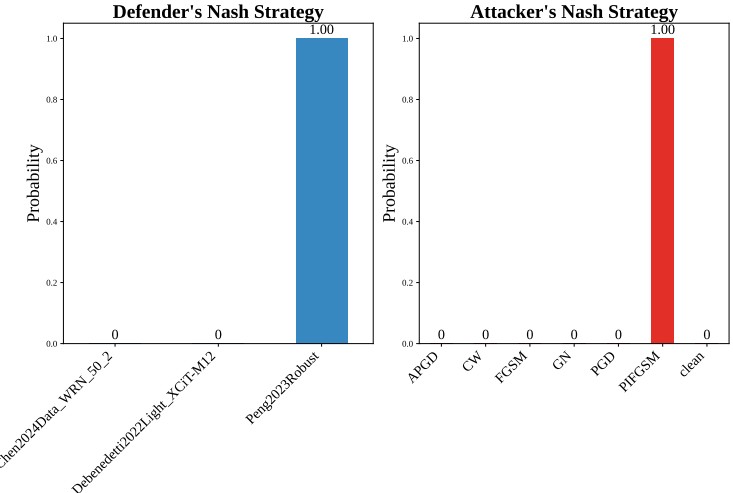

Figure 36: Obtain the Nash attack and Nash defense strategies on the ImageNet dataset using the TOPK method.

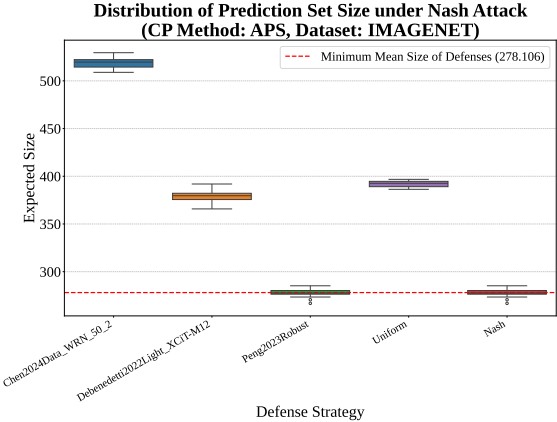

Figure 37: Box plots of size for different defense models using the APS method on the ImageNet dataset.

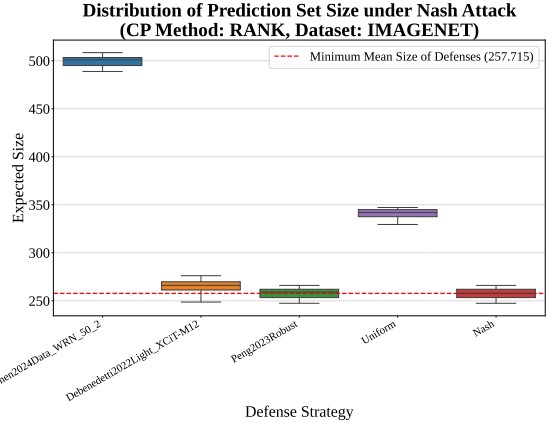

Figure 38: Box plots of size for different defense models using the RNAK method on the ImageNet dataset.

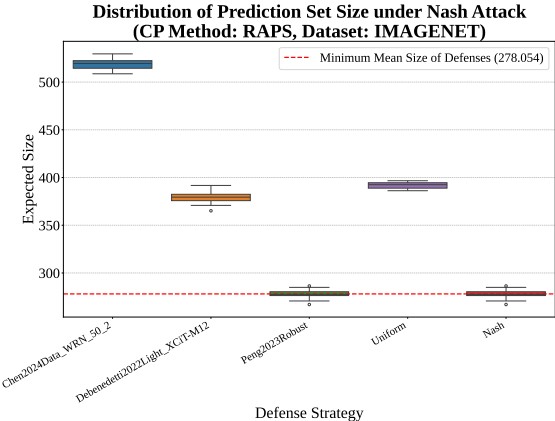

Figure 39: Box plots of size for different defense models using the RAPS method on the ImageNet dataset.

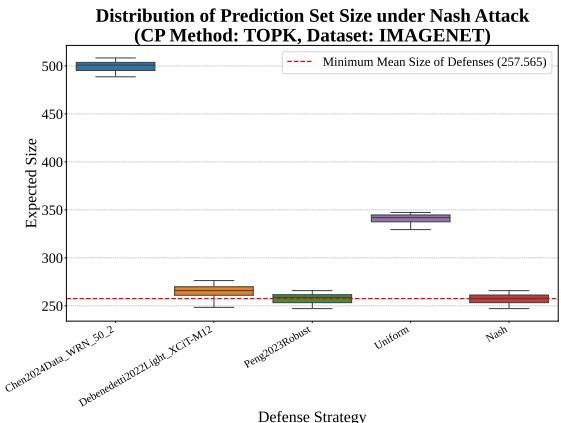

Figure 40: Box plots of size for different defense models using the TOPK method on the ImageNet dataset.

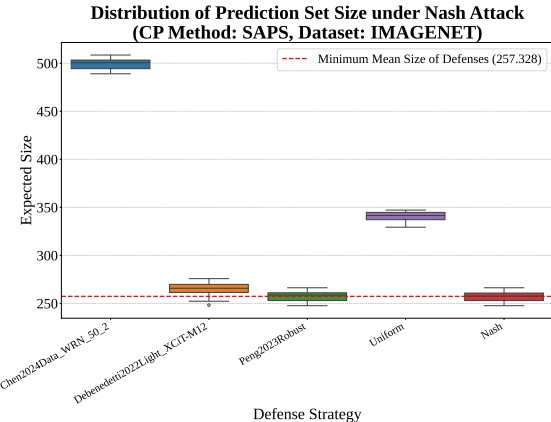

Figure 41: Box plots of size for different defense models using the SAPS method on the ImageNet dataset.

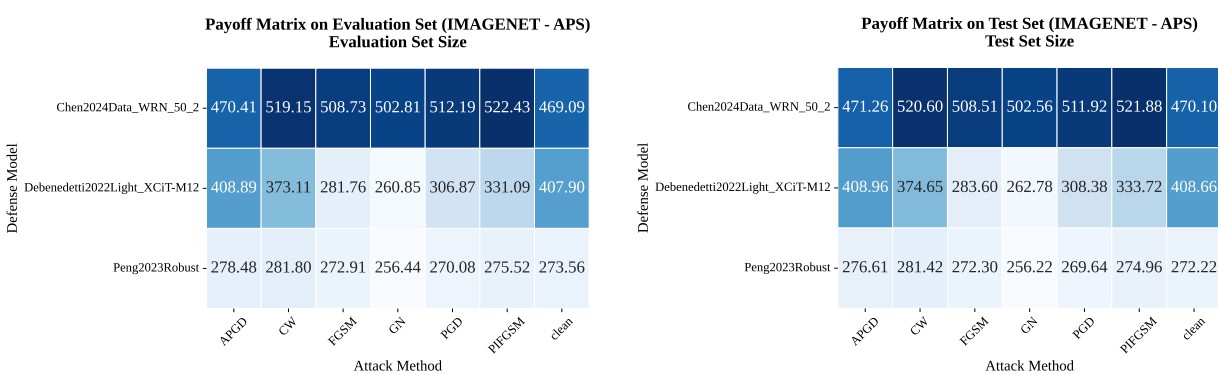

Figure 42: Evaluate the payoff matrix on both the ImageNet dataset's evaluation set and test set using the APS method.

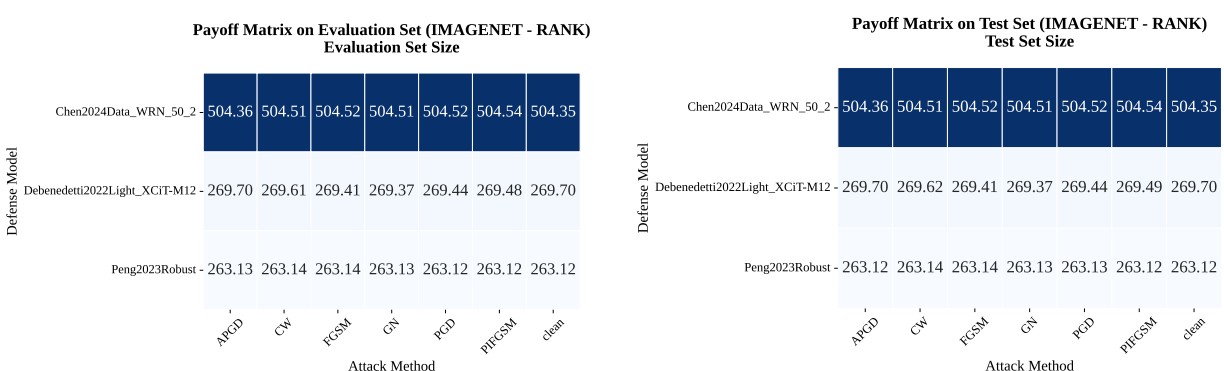

Figure 43: Evaluate the payoff matrix on both the ImageNet dataset's evaluation set and test set using the RANK method.

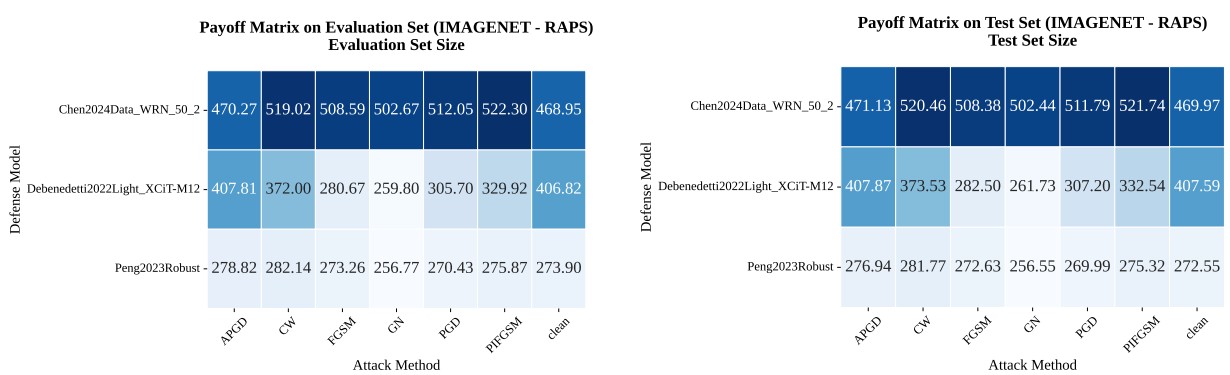

Figure 44: Evaluate the payoff matrix on both the ImageNet dataset's evaluation set and test set using the RAPS method.

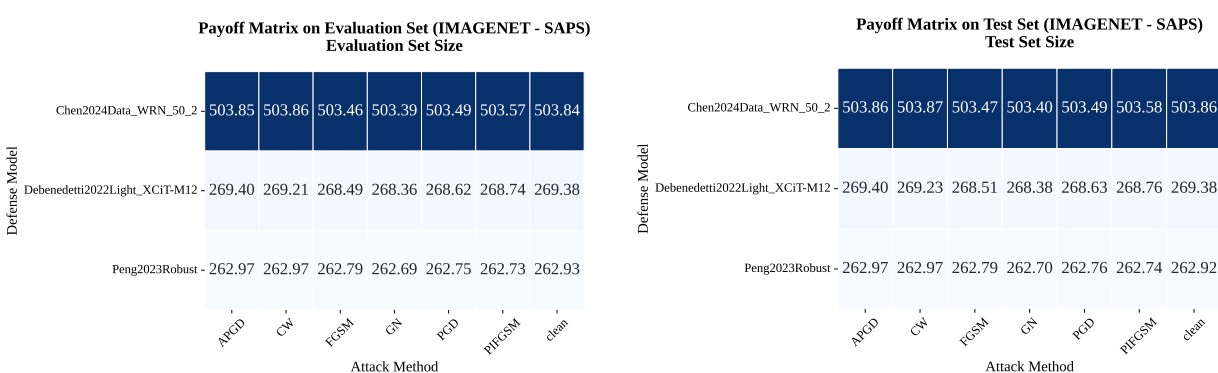

Figure 45: Evaluate the payoff matrix on both the ImageNet dataset's evaluation set and test set using the SAPS method.

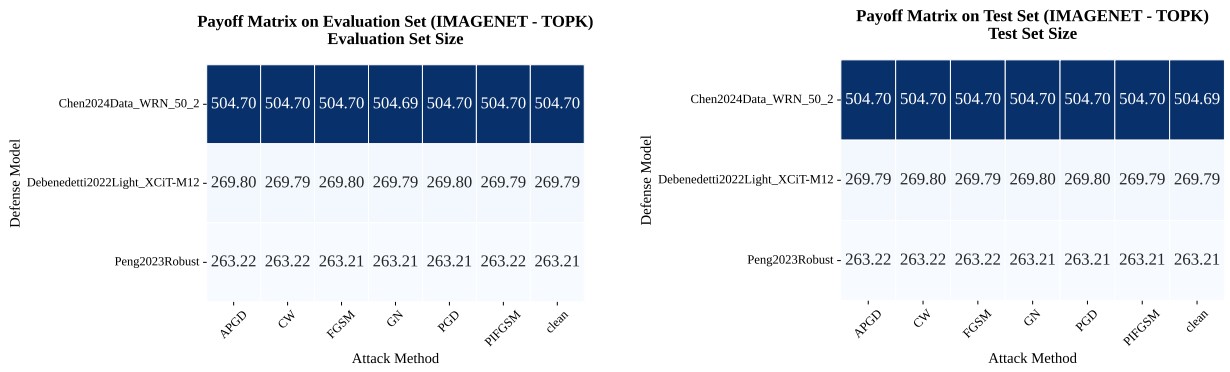

Figure 46: Evaluate the payoff matrix on both the ImageNet dataset's evaluation set and test set using the TOPK method.

