# OpenReview forum: "Game-Theoretic Defenses for Adversarially Robust Conformal Prediction"
_TMLR — Accepted by TMLR_

### Review · Reviewer_kaFT · 2025-10-28

**Summary Of Contributions:**

The authors propose a new model and methodology for training and evaluating deep learning models with conformal predictions robust against adversarial attacks. Deep learning models are known to be easily disturbed by adversarially constructed test input, and thus is often unreliable in safety-critical situations; traditional methods used to introduce robustness also suffers from a variety of drawbacks, including large complexity cost, inability to generalize to other adversarial attacks, etc.

To solve the aforementioned concerns, the authors employ game-theoretical ideas to train and select classifiers that serve as best strategies against a given set of adversarial attack strategies. By formulating the training-evaluating process as a two-player zero-sum game between the deep learning model as the defender and the adversary as the attacker, the authors propose a training paradigm involving solving for the Nash equilibrium and using the randomized strategy of selecting according to the optimal distribution on the family of available classifiers against the adversarial attack, provided that the strategy space for both the defender and the attacker is finite, i.e., the two-player zero-sum game in question is finite.

The authors provide experimental results evaluating the performance of their strategies on a variety of datasets, and across multiple attack/defense strategies.

While the authors' work points out a very promising and interesting direction for research, I am somewhat skeptical of the novelty and the extent of contribution of this submission. Despite reasonably pointing out some significant drawbacks of prior work on adversarially robust conformal predictions, the authors proceed to present a somewhat simple paradigm that does not seem to properly address these concerns, or still exhibits severe limitations, calling my doubts to the technical soundness and novelty of their claims.

**Audience:**

Yes

**Audience Explanation:**

I do believe that the (un)reliability of machine learning models in safety-critical scenarios is a very important research direction for researchers in the field, and is one that I am quite excited for. Despite the submission's potential lack of novelty, I think the topic of choice and the game-theoretical direction that the authors propose is indeed informative for the audience of TMLR as a whole.

**Broader Impact Concerns:**

I do not believe a broader impact statement is required concerning the nature of this work.

**Claims And Evidence:**

No

**Claims Explanation:**

Expanding on my opinions in the summary section: I find the authors' claims on the novelty and contribution of their game-theoretical paradigm rather insufficiently substantiated for a large part.

The authors' claims, in my understanding, are as follows:
- Prior works crucially rely on the exchangeability assumption, which is difficult to satisfy in real-world and adversarial scenarios;
- Prior methods to introduce robustness to conformal predictions suffer from an inability to generalize against unknown attack strategies.
Both of which the authors claim to address with their training paradigm.

However, none of these crucial points seem to be properly and completely addressed by this submission:
- The algorithmic framework on page 5 trains, calibrates, and evaluates exclusively on the training data, split into three partitions. The testing data does not seem to play any part before the final classifier-strategy is selected. Intuitively, this calls for the same concerns as prior training paradigms - that should the training set prove insufficiently representing, the optimal strategy can still perform suboptimally on unknown data.
- The authors' two-player zero-sum game theoretic formulation continues to rely on the assumption that the strategy space for both the model and the adversary are finite, which is a very limiting assumption. When presented with unknown attack strategies outside of the considered attack strategy space, the theoretical guarantees obtained by the framework can fall apart similar to prior works.
The submission does not seem to address these observable drawbacks, and from the current manuscript I cannot find confidence in the authors' claims of novelty and contribution.

**Requested Changes:**

As illustrated above, I am unsatisfied with the authors' claim of contribution and novelty; I believe the limitations and drawbacks of their training paradigm and framework should be properly addressed and discussed before it can be properly re-evaluated for acceptance.

---

> ### Author Response · Authors · 2025-11-23
>
> We thank the reviewer for reviewing our paper. We notice the reservations regarding novelty and the questions about our methodology. We hope to clarify these misunderstandings and articulate the value of our work below.
>
> **1. Response to "Simple Paradigm" and Novelty**
>
> We respectfully clarify that the "simplicity" of our approach is a deliberate design choice to ensure **verifiability** and **generality**.
> *   Existing adversarial CP methods (e.g., RSCP) often rely on specific smoothing techniques or heuristic defenses.
> *   Our contribution is a game-theoretic framework that provides a **provable lower bound guarantee** on performance. Our framework captures the intrinsic adversarial dynamics, allowing the defender to optimize a randomized strategy that remains robust even when the attacker plays their optimal counter-strategy.
> *   By decoupling the defense strategy from the underlying CP scoring function, our paradigm allows for the integration of any state-of-the-art defense model and conformal prediction method while providing a mathematically verifiable worst-case bound via Nash Equilibrium.
>
> **2. Clarification on Data Splits and Algorithm Correction**
>
> *   **Purpose of Splits:** The *Validation/Calibration set* is used strictly to solve for the Nash Equilibrium strategy (finding the optimal mixing weights). The *Test set* is used solely to evaluate the performance of this Nash defense against various attacks.
> *   **Correction:** We apologize for the confusion caused by the presentation in the previous version. We have corrected **Algorithm 1**, specifically separating the description of Steps 4 and 5, which were erroneously merged.
> *   **Figure 2:** As noted, Figure 2 is intended to verify the **exchangeability** between the validation and test sets. The results show that the prediction set sizes generated by the same model under the same attack are highly consistent across both sets, confirming that the Nash strategy derived from the validation set is valid and effective on the test set.
>
> **3. Limitation of Finite Strategy Space**
>
> We acknowledge this limitation and have discussed it as a direction for future work. However, we emphasize two points:
> 1.  **Challenge:** Obtaining non-asymptotic guarantees for infinite/continuous attack spaces is an open challenge in the field.
> 2.  **Generalization:** To address this, we added "Leave-One-Out" experiments (Appendix A.5). The results show that our method generalizes to unknown attacks. While no method can guarantee coverage against *arbitrary* unknown attacks, formulating the problem as a finite game allows us to efficiently solve for a Nash Equilibrium using Linear Programming. This provides a robust lower bound against known high-risk threats, which is highly valuable for safety-critical applications (e.g., autonomous driving) where specific attack types are the primary concern.

---

### Review · Reviewer_g7zA · 2025-11-04

**Summary Of Contributions:**

### Strengths
- The paper formalizes the problem of selecting a robust defense model for Conformal Prediction as a two-player, zero-sum game from a finite, predefined set of attackers and defenders.
- The paper presents a fairly straightforward motivation, and I truly appreciated the discussion in Section 2 for clarity and formalization of Conformal Prediction. The paper was also relatively easy to understand.

---

### Weaknesses / Questions
- The theorem stated in the paper and nash equilibrium holds true only if the attacker attacks from a known set of attacks. The issue is attackers often pick attacks which are unknown - which breaks the guarantee / size of the prediction set. Can the authors provide experiments for how the Nash defense holds up against an unknown style of attacks?
- Why do the authors split the dataset into train and calibration? Why not use only one dataset?
- Can you explain why Chen-2024-Data consistently does well in Figures 4 and 5?

**Audience:**

Yes

**Audience Explanation:**

Yes, these findings would definitely be interesting.

**Broader Impact Concerns:**

Not applicable.

**Claims And Evidence:**

Yes

**Claims Explanation:**

Yes, majority of the claims seems to be supported by clear evidence. However, a few major questions remain underanswered as shown in the Weaknesses.

**Requested Changes:**

Critical changes are already discussed in Weaknesses. Regardless, there are some minor revisions that need to be made:

- If the captions to each figure (especially to those in the Appendix) were more informative, it would be appreciated.
- The paper’s citations are not properly formatted, which significantly impacts readability. We kindly request the authors to address this issue to enhance the presentation quality.

---

> ### Author Response · Authors · 2025-11-23
>
> Thank you for your affirmation of our work, especially for appreciating the clarity in Section 2. In response to the weaknesses and questions you've raised, our replies are as follows:
>
> **1. Performance under Unknown Attacks**
>
> (Please refer to our response to Reviewer 1 regarding the new "Leave-One-Out" experiments in Appendix A.5). We have empirically demonstrated that our Nash-based strategy generalizes well to unseen attacks, provided the known attack portfolio is sufficiently strong.
>
> **2. Why separate Calibration and Training sets?**
>
> This is a fundamental requirement of Conformal Prediction to ensure **exchangeability**.
> *   If we calculate non-conformity scores on the **training set**, the scores would be artificially low due to model overfitting.
> *   Consequently, the resulting threshold ($q_{1-\alpha}$) would be too small.
> *   When applied to unseen test data (where errors are naturally higher), this low threshold would fail to cover the true labels, violating the coverage guarantee.
> *   Therefore, using a held-out **Calibration Set** is essential to obtain an unbiased estimate of the non-conformity score distribution.
>
> **3. Reasons for the superior performance of Chen-2024**
>
> We appreciate this observation. Based on our analysis of Chen et al. (2024), we attribute its superior performance to two factors:
> *   **High Natural Accuracy & Robustness:** Unlike standard adversarial training that often trades natural accuracy for robustness, Chen-2024 employs a dynamic data filtering strategy to remove inefficient adversarial examples during training. This allows the model to maintain high confidence on true labels (high natural accuracy) while being resilient to perturbations.
> *   **Tighter Non-Conformity Scores:** In the context of Conformal Prediction, a model that is both accurate and robust yields lower non-conformity scores across the board. This results in a smaller $q _{1-\alpha}$ threshold compared to models that suffer from accuracy degradation, allowing the Chen-2024 backbone to produce more efficient (smaller) prediction sets without violating the coverage guarantee.
>
> **4. Figure Captions and References**
>
> We thank the reviewer for the detailed check. We have rewritten the figure and table captions (especially in the Appendix) to be more descriptive and self-explanatory. We have also thoroughly formatted all references to comply with TMLR standards to improve readability.

---

### Review · Reviewer_phk5 · 2025-11-20

**Summary Of Contributions:**

This paper tackles the challenge of obtaining reliable uncertainty quantification under adversarial attacks, which frequently break the exchangeability assumptions required for conformal prediction. The authors introduce a framework that integrates conformal prediction with a discrete zero-sum game between an attacker and a defender. By analyzing this interaction, they derive a Nash equilibrium defense strategy that is proven to maintain valid coverage while minimizing worst-case prediction-set size within a predefined attack space. Experiments show that the defense consistently achieves both the desired coverage and smaller, more stable prediction sets

**Audience:**

Yes

**Audience Explanation:**

The paper connects adversarial robustness and conformal prediction, two active research areas in machine learning.

The game-theoretic formulation would interest researchers in reliable, uncertainty quantification, and robust deep learning.

Its practical relevance to safety-critical domains (such as autonomous driving and medical imaging) makes it appealing to a wide range of applications.

**Claims And Evidence:**

Yes

**Claims Explanation:**

The theoretical claim about preserving coverage using max-quantiles is correct and clearly justified.

The proposed method and its formulations are correct.

Experiments on CIFAR-10/100 and ImageNet support the claims

**Requested Changes:**

1. The current framework guarantees coverage only for attacks included in the predefined attack set. Adding experiments or analysis on attacks not in the calibration set would strengthen confidence in the method’s practical robustness.

2. Discuss assumptions on known finite attack space. Since the main theoretical guarantee relies on this assumption, adding a brief discussion of limitations and potential extensions would improve clarity.

3. Provide more intuition behind the max-quantile guarantee. A short intuitive explanation or toy example could help readers unfamiliar with conformal prediction understand why the guarantee holds.

---

> ### Author Response · Authors · 2025-11-23
>
> Thank you for your positive feedback on our work. We are greatly encouraged by your recognition of our theoretical proposition regarding the use of max-quantiles to maintain coverage, as well as the formalization of Nash equilibrium. In response to your constructive amendments and questions, our replies are as follows:
>
> **1. Experiments regarding Unknown Attacks**
>
> We appreciate this suggestion. To evaluate the generalization capability of our framework against attacks not included in the pre-defined set, we have added extensive experiments in **Appendix A.5 (Tables 2 & 3)**.
>
> *   **Methodology:** We employed "Leave-One-Out" and "Leave-Two-Out" setups. Specifically, we excluded one or two types of attacks from the game formulation (Nash strategy construction) and used these excluded attacks solely during the testing phase to evaluate the defense.
> *   **Results:** The results demonstrate that the Nash defense strategy maintains high coverage and reasonable prediction set sizes even against unseen attacks, proving the framework's generalized robustness.
> *   **Observation:** We candidly observe that if an unknown attack is significantly stronger than any attack in our portfolio, the theoretical coverage guarantee may not hold. However, our experiments suggest that constructing a game with a sufficiently diverse set of strong attacks effectively mitigates the risk of most "out-of-distribution" attacks.
>
> **2. Assumption of Finite Attack Space**
>
> We fully agree with the reviewer. The current theoretical guarantee relies on the assumption of a finite attack space. We have added a dedicated paragraph in the **Conclusion** section to explicitly state this limitation. We clarify that while real-world attack spaces are continuous, modeling them as a discrete game is a necessary step toward verifiable defense. Furthermore, by covering the most dominant attack paradigms, we can approximate the worst-case threats in practical applications.
>
> **3. Intuition behind Max-Quantile Guarantee**
>
> We have added a **Remark in Section 3.1** to specifically discuss the intuition of max-quantiles in Conformal Prediction. We explain that by calibrating to the "worst-case" quantile among known attacks, we inherently satisfy the coverage requirements for weaker or similar attacks.

---

### Decision · Action_Editor_BiH7 · 2026-01-05

**Recommendation:** Accept as is

**Additional Comments:**

This paper addresses the challenge of reliable uncertainty quantification under adversarial attacks, which often violate the exchangeability assumptions required by conformal prediction. The authors propose a novel framework that formulates the interaction between an attacker and a defender as a discrete zero-sum game integrated with conformal prediction.

The reviewers all support acceptance of this paper, due to the novelty idea and promising empirical results. However, the theorectical analysis is a little bit rough and can be further improved, which can be a separate work. Based on the review comments, I recommend acceptance of this paper.

**Audience:**

Yes

**Audience Explanation:**

The researchers working on adversarial attacks and uncertainty quantification can be potential audience of this paper

**Claims And Evidence:**

Yes

**Claims Explanation:**

The proposed approach has been demonstrated to be effective by theoretical analysis and experiments on several image datasets.